# A functional screen uncovers circular RNAs regulating excitatory synaptogenesis in hippocampal neurons

Darren Kelly[1], Silvia Bicker[1], Jochen Winterer [1], Prakruti Nanda[1], Pierre-Luc Germain [1,2,3], Christoph Dieterich [4] & Gerhard Schratt [1] ✉

Circular RNAs (circRNAs) are an expanding class of largely unexplored RNAs which are prominently enriched in the mammalian brain. Here, we systematically interrogate their role in excitatory synaptogenesis of rat hippocampal neurons using RNA interference. Thereby, we identify seven circRNAs as negative regulators of excitatory synapse formation, many of which contain high-affinity microRNA binding sites. Knockdown of one of these candidates, *circRERE*, promotes the formation of electrophysiologically silent synapses. Mechanistically, *circRERE* knockdown results in a preferential upregulation of synaptic mRNAs containing binding sites for miR-128-3p. Overexpression of *circRERE* stabilizes miR-128-3p and rescues exaggerated synapse formation upon *circRERE* knockdown in a miR-128-3p binding site-specific manner. Overall, our results uncover *circRERE*-mediated stabilization of miR-128-3p as a means to restrict the formation of silent excitatory synaptic co-clusters and more generally implicate circRNA-dependent microRNA regulation in the control of synapse development and function.

Non-coding RNAs (ncRNAs) play increasingly appreciated gene-regulatory roles in the brain[1] and can exist in both linear and circular forms. Circular RNAs (circRNAs) are characterized by their stability and closed ends, which are formed through a back-splicing event in which a covalent bond forms between a pre-mRNA 5′ splice donor and a 3′ splice acceptor site, leading to the formation of a unique backsplice junction sequence (BSJ)[2]. Recent work has further identified the dedicated mechanism by which circRNAs are exported to the cytoplasm dependent upon Ran-GTP/Exportin-2, underscoring the potential relevance of circRNA localization for cell function[3]. The circRNA transcriptome in the nervous system is particularly diverse and abundant[4]. circRNAs are highly expressed in developing neurons, often localized to the synapto-dendritic compartment, and many are derived from genes with synaptic functions[5]. Moreover, circRNAs are

regulated in a neuronal activity-dependent manner[6], together suggesting that they could be involved in the control of synaptogenesis and plasticity. However, the function of the vast majority of the hundreds of conserved circRNAs expressed in mammalian neurons is still unexplored.

Arguably the most intensely studied neuronal circRNA is complementary-determining region 1 antisense (*Cdr1-as*). *Cdr1-as* contains more than 70 binding sites for the neuronal microRNA (miRNA) miR-7 and acts as a miR-7 sponge[7], as part of a complex non-coding regulatory network[8,9]. The loss of *Cdr1-as* has been shown to lead to miR-7 destabilization in a neuronal activity-dependent manner, thereby affecting excitatory synaptic transmission and schizophrenia-associated behavior[8,10]. Moreover, localized *Cdr1-as* activity is required for fear memory extinction[11]. In addition to *Cdr1-as*, a few additional

[1]Laboratory of Systems Neuroscience, Institute for Neuroscience, Department of Health Science and Technology, ETH Zürich, Zurich, Switzerland. [2]Laboratory of Molecular and Behavioural Neuroscience, Institute for Neuroscience, Department of Health Science and Technology, ETH Zürich, Zurich, Switzerland. [3]Lab of Statistical Bioinformatics, IMLS, University of Zürich, Zurich, Switzerland. [4]Section of Bioinformatics and Systems Cardiology, Department of Internal Medicine III and Klaus Tschira Institute for Integrative Computational Cardiology, University of Heidelberg, Heidelberg, Germany. ✉e-mail: Gerhard.schratt@hest.ethz.ch

circRNAs (e.g., *circSatb1, circFat3, circGRIA1, circHomer1*) have been implicated in the regulation of synapse development, plasticity, and cognition[12–16].

Besides their role in regulating miRNA activity[7], circRNAs have been shown to act through a variety of mechanisms including transcription and splicing regulation in the nucleus[17], RBP interaction[18], and translation[19]. Recently, neuronal circRNAs have been further implicated in neurological and neuropsychiatric diseases[20]. For example, *circIgfbp2* is significantly induced by traumatic brain injury (TBI), which is associated with several neurological and psychiatric disorders, and *circIgfbp2* knockdown (KD) alleviated mitochondrial dysfunction and oxidative stress-induced synaptic dysfunction after TBI[21]. *circHomer1* is downregulated in the prefrontal cortex of bipolar disorder and schizophrenia patients, and *circHomer1* KD in the mouse orbitofrontal cortex results in deficits in cognitive flexibility[15]. Furthermore, circRNAs are differentially expressed in pre- and post-symptomatic Alzheimer's[22] and Huntington's disease[23].

CircRNAs are thus implicated in the regulation of synaptic plasticity and control of protein synthesis with relevance in the pathogenesis of neurodevelopmental and psychiatric disorders. However, the vast numbers, tissue specificity, and splicing isoform complexity of circRNAs make it difficult to distinguish those with a functional role from a majority produced as presumably non-functional splice variants[24].

In this work, we investigate circRNA function in mammalian synaptogenesis in a more systematic manner, by first defining the circRNA landscape in neuronal processes (axons and dendrites), followed by RNA interference (RNAi)-mediated knockdown of process-enriched circRNAs. This approach identifies several candidate circRNAs for detailed mechanistic and functional investigation in follow-up experiments.

## Results

### Characterization of the circRNA landscape in the process compartment of primary rat hippocampal neurons

We reasoned that circRNAs which are highly abundant in neuronal processes (i.e., axons and dendrites) might represent strong candidates for circRNAs with functional relevance in synaptogenesis. To identify this neuronal circRNA population, we employed a previously characterized compartmentalized rat hippocampal neuron culture system, in which the transcriptome of neuronal somata and processes can be individually analyzed after physical separation[25]. By re-analyzing our published ribo-depleted transcriptomic dataset[26], followed by circRNA reconstruction of unique BSJs[27] we detected 1041 unique circRNAs, of which 907 circRNAs could be detected in the somata compartment and 1027 in processes (Fig. 1A, Supplementary Table 1).

circRNA-specific BSJ reads displayed a strong process compartment enrichment overall relative to the somata compartment when normalized to the total transcriptome (Fig. 1B; 234 circRNAs display a significant enrichment at FDR < 0.05). This systematic process enrichment of circRNAs is also reflected upon normalization to the total reads of the circular RNA + linear RNA between the BSJ loci (Fig. 1C). This data suggests that neuronal circRNAs, in general, are biased toward localization to neuronal processes, strongly implicating them in the regulation of local gene expression during synaptogenesis. However, circRNA process enrichment likely occurs in a sequence-specific manner[11], since several circRNAs were either not significantly enriched in the process compartment (e.g., *Cdr1-as*) or even displayed strong soma enrichment (e.g., *circSnap25*) (Fig. 1D). We went on to validate selected process-enriched circRNAs using our previously established single-molecule fluorescent in situ hybridization (smFISH) protocol[28]. Thereby, we were able to confirm dendritic localization of the circRNAs *circHomer1, circStau2*, and *circRMST* in hippocampal pyramidal neurons (Fig. 1E). The circularity of selected circRNAs was

further validated by qPCR upon RNase R treatment (Supplementary Fig. 1A, B).

Next, we decided to analyze the genetic origin of process-enriched circRNAs in further detail. Gene set enrichment analysis of circRNA host genes shows that process-enriched circRNAs are disproportionately often produced from synaptic genes, particularly those encoding postsynaptic proteins (Fig. 1F). CircRNAs that are enriched in the process compartment are 4.2 times more likely to be on the SFARI autism risk gene set (65/234 circRNAs, $p < 2e-19$, two-sided Fisher's exact test) (Supplementary Table 2). Together, our bioinformatics results are consistent with a role for process-enriched circRNAs in synaptogenesis and indicate a potential link to synaptopathies, such as autism spectrum disorders.

Most of the published examples of neuronal circRNAs act via miRNA association. However, miRNA sites in general do not appear to be overrepresented in circRNAs[29]. In agreement with the latter result, circRNAs which are expressed in our neuron culture system on average do not contain more miRNA binding sites in comparison to mRNA 3′UTRs or whole transcripts (Supplementary Fig. 2A–C). However, miRNA binding sites are slightly more abundant in process-enriched circRNAs compared to the total population ($P = 0.044$) (Supplementary Fig. 2D). This effect is less pronounced when only considering 8mer binding sites or when normalizing to the sequence length (Supplementary Fig. 2E, F). A higher abundance of miRNA sites appears to be restricted to a subset of circRNAs which overall contain a large number of binding sites (Supplementary Fig. 2D, E), suggesting the presence of a specific circRNA sub-population with a high potential for miRNA regulation. Specific examples of process-enriched circRNAs containing many predicted miRNA binding sites are shown in Supplementary Fig. 2G, H. Taken together, processes of primary rat hippocampal neurons contain hundreds of circRNAs whose function in neuronal development is unexplored.

### RNAi screen reveals process-enriched circRNAs with a function in excitatory synaptic co-cluster and dendritic spine formation

Based on our results from RNA sequencing, we decided to systematically characterize the function of process-enriched circRNAs in excitatory synaptogenesis. To narrow down the list of candidates to a number that can be handled in the context of a screen in primary neurons, we applied further selection criteria in addition to process enrichment, namely absolute expression levels in processes (>10 counts in RNA-seq) and sequence conservation (orthologues in mouse and human). This procedure led to the selection of 30 candidate circRNAs. Sometimes, this led to the inclusion of multiple isoforms from the same gene locus, e.g., in the case of *circRERE, circANKS1B*, and *circRMST* (Fig. 2A). Additionally, the published miRNA regulatory circRNAs *Cdr1-as*[7] and *circHipk3*[30], were added to the screen.

siRNA pools (three independent siRNAs targeting the unique BSJ) were generated for each of the 32 circRNAs, as well as purchased two commercially unrelated scrambled sequences for control purposes (siControl1, siControl2). Primary hippocampal neuronal cultures (DIV8) were transfected with GFP plasmid and siRNA pools followed by immunocytochemistry for synapsin-1/PSD-95 at DIV16, a time when excitatory synapse formation peaks in primary hippocampal neuron cultures. Subsequently, Synapsin-1/PSD-95 co-cluster density, which serves as a proxy for excitatory synapse density[31], was automatically determined with a FIJI pipeline using GFP as a cell mask, as also described previously[32]. In total, 7/32 siRNA pools against specific circRNAs resulted in a significant increase in synapse density, suggesting that these circRNAs act as endogenous repressors of synapse formation and/or maintenance (Fig. 2B–D; Supplementary Fig. 3A). Importantly, increased synapse density for these candidates was observed

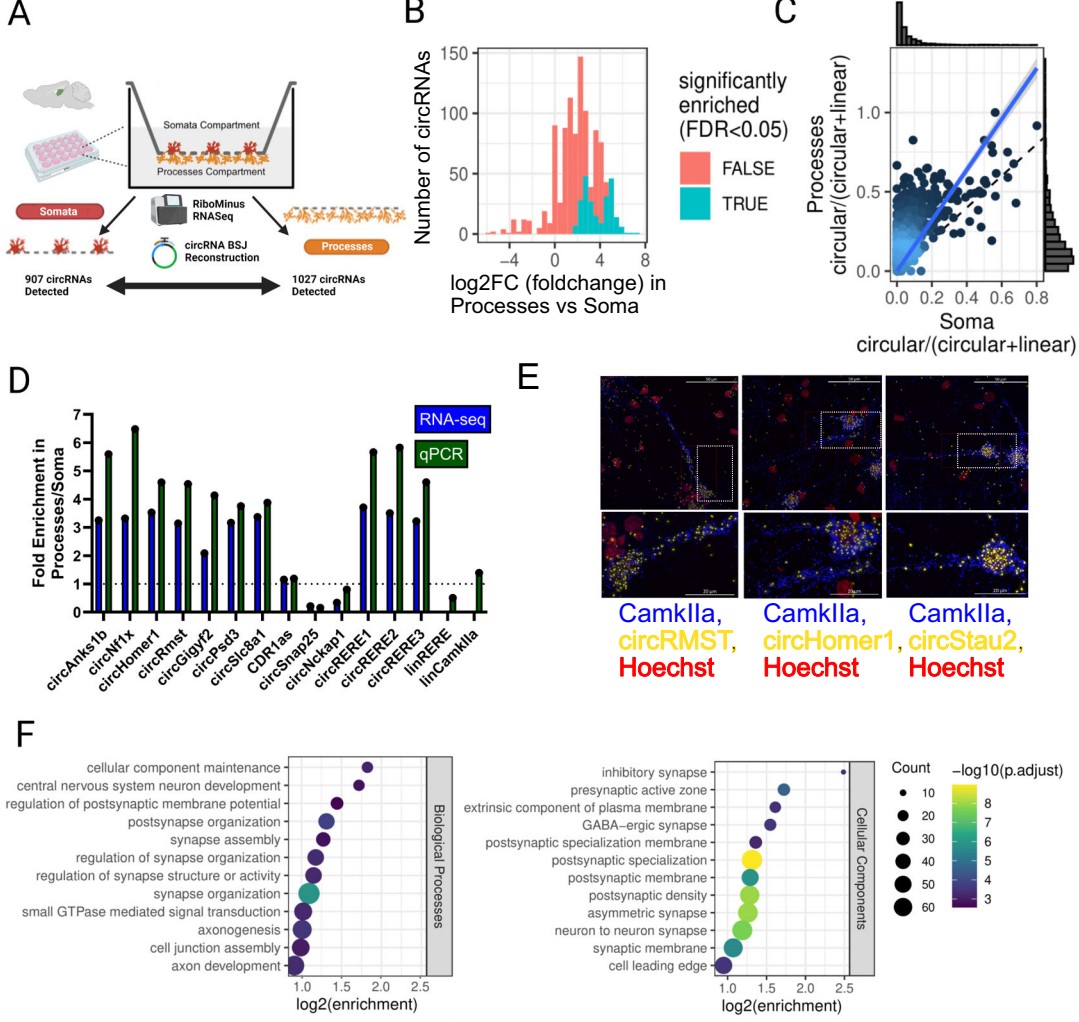

**Fig. 1 | The circRNA landscape in processes of rat hippocampal neurons.**
**A** Schematic of workflow for the profiling of circRNAs from compartmentalized rat hippocampal neurons by RiboMinus RNA-seq. circRNA BSJs were reconstructed from sequencing (DCC/Ciri) and quantified. **B** Histogram of circRNA BSJ enrichment in the process compartment normalized to the total transcriptome, 4 replicates per group, 234 circRNAs showed significant enrichment in the process compartment (blue; RNA-seq DEA; FDR < 0.05). **C** circular/(circular + linear) ratios of circRNA-producing genes in processes and soma compartment, 4 replicates per group. Enrichments for circular RNAs are heavily skewed toward the process compartment ($p < 2.2e\text{-}16$, 2-sided Wilcoxon Test). **D** RT-qPCR with divergent

primers for selected circRNA candidates. circRNA enrichment (normalized to Gapdh mRNA) in processes/soma is shown in comparison to enrichments obtained by RNA-seq (**B**). **E** smFISH of selected circRNA candidates with a probe designed against BSJ; Camk2a mRNA probe as a neuronal marker ($N = 1$, 5 neurons imaged). **F** GO overrepresentation analysis (i.e., one-sided) for host genes of process-enriched circRNAs, using Biological Processes and Cellular Components ontologies, performed using clusterProfiler 4.10.1 using the genes passing DEA-filtering as background and BH multiple-testing correction. Created in BioRender. https://BioRender.com/g74b060. Source data are provided as a Source Data file.

when using independent siControls for comparison (Supplementary Table 3), illustrating high robustness of our findings.

To obtain independent support for the role of circRNAs in excitatory synapse development, we performed a secondary screen using as a read-out dendritic spines, the major postsynaptic sites of excitatory synaptic contact. In this analysis, 11/32 candidates showed increased dendritic spine density when compared to siControl conditions (Supplementary Fig. 3B; Supplementary Table 3), with siRNA pools against *circRERE2* (official nomenclature: *circRERE* (5,6,7,8,9,10,11)) and *circPhf21a* being the only conditions displaying a robust increase in both spine density and synapse co-cluster density. When analyzing dendritic spine volume (Supplementary Fig. 3C; Supplementary Table 3), the only circRNA that resulted in a phenotype was *Cdr1-as*, whose knockdown significantly reduced dendritic spine size. In contrast, transfection of siRNA pools against *circRERE2* and *circPhf21a* had no significant effect on dendritic spine volume, suggesting

a specific effect on excitatory synapse formation. In conclusion, results from our siRNA screen suggest that several process-enriched neuronal circRNAs control different aspects of excitatory synapse development. Among those, *circRERE2* and *circPhf21a* represent particularly strong candidates since they affect both the formation of excitatory synaptic co-clusters and dendritic spines.

*circPhf21a* has been previously identified as a highly expressed circRNA (2nd most highly expressed circRNA in the rat brain, circAtlas 3.0[33]). It is spliced from the 5′ UTR of *Phf21a* pre-mRNA and is enriched in the brain and in synaptoneurosomes[5]. In contrast, much less is known regarding the neuronal function of *circRERE*. Intriguingly, the knockdown of a second *circRERE* splice isoform, *circRERE 1* (official nomenclature: *circRERE* (5,6,7,8,9,10)), similarly resulted in an, albeit non-significant, increase in the synapse ($P < 0.14$) and spine density ($P < 0.19$) (Fig. 2C). We, therefore, decided to focus on *circRERE* isoforms for our further mechanistic studies.

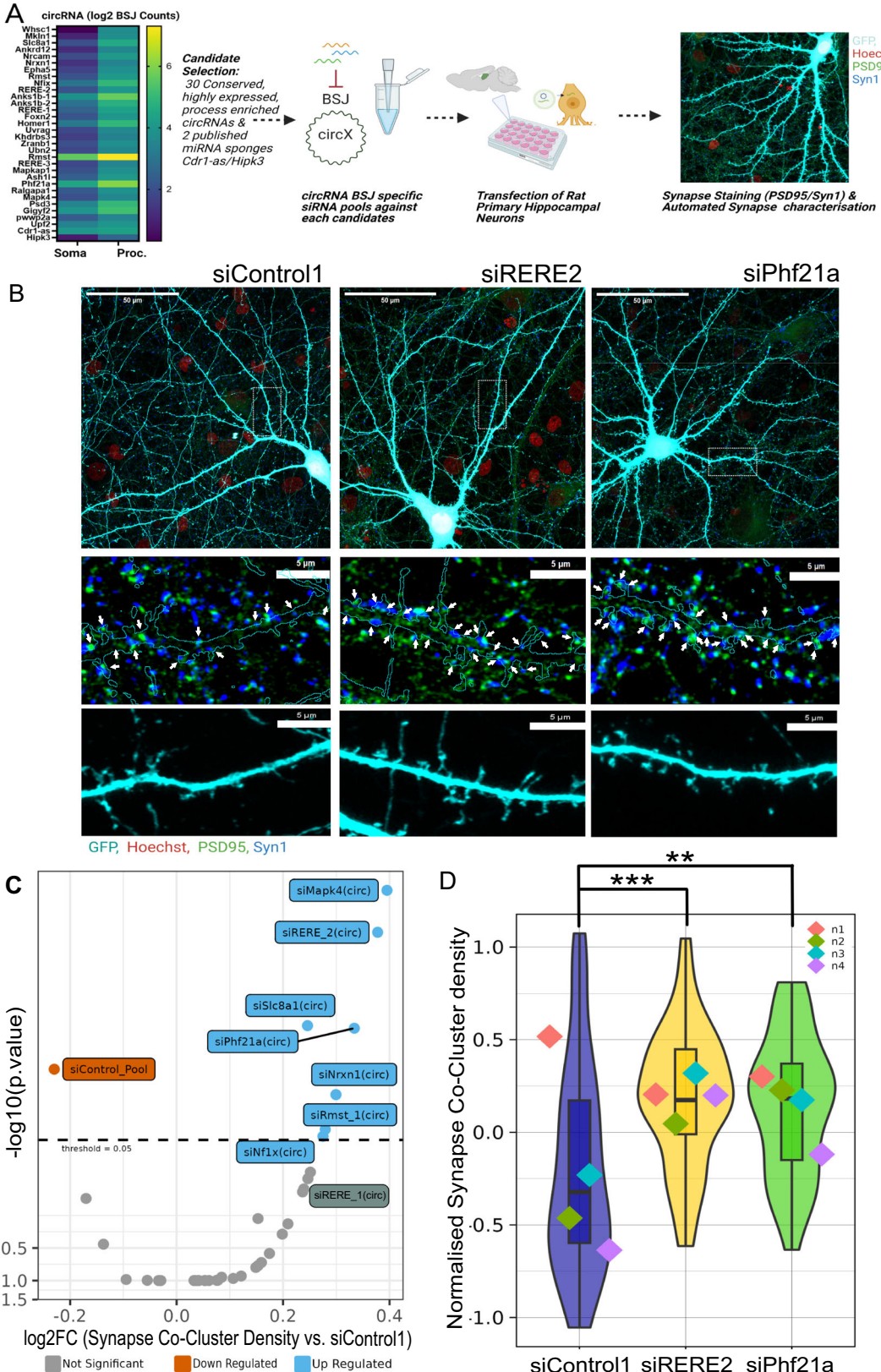

**circular RERE isoforms suppress the formation but promote the functional maturation of excitatory synaptic co-clusters**

The *RERE* locus produces multiple *circRERE* isoforms due to different back-splicing events (Fig. 3A). This feature is evolutionarily conserved, with illustrated comparisons from rat (this study), mouse[11], and human *circRERE*[34]. Three isoforms (*circRERE 1*, *circRERE2* and *circRERE3*)

(official nomenclature: *circRERE* (5, 6, 7, 8, 9, 10, 11, 12)), which are back-spliced out of exons 5–10/11/12 of the *RERE* gene, are the most expressed, highly process-enriched *circRERE* isoforms (Fig. 3A, B). circAtlas 3.0 data repository counts *circRERE 1* (circAtlas ID: rno-Rere_0005) and *circRERE2* (circAtlas ID: rno-Rere_0019) as the 13th and 28th most highly expressed circRNAs in rat brain, respectively[33].

**Fig. 2 | Several process-enriched circRNAs inhibit the formation of excitatory synaptic co-clusters and dendritic spines. A** Workflow of the circRNA functional screen. Heatmap of candidate circRNAs (30 most process-enriched circRNAs plus *Cdr1-as* and *Hipk3*) for which siRNA pools (3 independent siRNAs per circRNA) were designed. Immunocytochemistry for pre (synapsin-1) and postsynaptic (PSD-95) markers was performed at DIV16, followed by an automated count of synaptic co-clusters based on a GFP mask which labels transfected cells. **B** Representative Images of siControl condition as well as two selected circRNA candidates (si-*circPhf21a*, si-*circRERE2*) which display increased synaptic density upon knockdown. Red: Hoechst; green: PSD-95; blue: Synapsin-1; turquoise: GFP. Upper panel: overview images. Middle and lower panel: Magnification of boxed insets depicting either synaptic co-clusters (middle, marked by arrows) or dendritic segments with spines (lower). 8 images were taken (i.e., cells per condition per replicate). **C** Volcano Plot showing $\log_2$ fold-changes in synapse co-cluster density (x-axis) vs. $-\log_{10}p$-value (y-axis) for individual siRNA pools directed against circRNA candidates (two-sided). Significant up-or downregulated siRNA pools ($p < 0.05$), as well as siRERE_1 ($p < 0.14$), are labeled. **D** Synapse co-cluster density (normalized to the GFP-only condition present on the same plate) for siRNA pools directed against indicated circRNA candidates (two-sided). Violin plots with embedded boxplots, including the median, interquartile range, and whiskers from minimum to maximum. Pairwise comparisons (emmeans/GLMM): siRERE_2 vs. siControl1: logFC = 0.37764 ± 0.0874, $t$-ratio = 4.32097, $p = 0.000582117$. siPhf21a vs. siControl2: logFC = 0.33387 ± 0.0874, $t$-ratio = 3.82025, $p = 0.0045$. $P < 0.001 = ***, P < 0.01 = **$. C/D: GLMM statistical modeling, $n = 4$ independent biological replicates, 8 cells per condition. All comparisons are provided in Supplemental Table 3. Created in BioRender. https://BioRender.com/d06j864. Source data are provided as a Source Data file.

Transcriptomic analysis of all *RERE*-related transcripts demonstrates an overall process enrichment, supporting *circRERE* isoforms to be the predominant expressed RNA type in neurons when compared to linear *RERE* mRNA (Supplementary Fig. 4A). *circRERE* 1, 2, and 3 isoforms were also shown to be circular by resistance to RNase R degradation, in contrast to *linRERE* (Supplementary Fig. 1B).

Circular and linear *RERE* isoforms are relatively uniformly expressed in different rat brain regions (Fig. 3C). Furthermore, *circRERE* and linear *RERE* mRNA share comparable expression patterns in human iPSC neurons (Supplementary Fig. 4B, C)[34], with circular isoforms corresponding to rodent *circRERE 1* and *circRERE2* being dominant circular isoforms. We went on to characterize the most abundant *circRERE* isoform, *circRERE2*, in further detail. Rolling circle amplification[35] of *circRERE2* utilizing BSJ-spanning divergent primers demonstrates its circularity and confirms the expected size of 691nt size (Supplementary Fig. 4D). We performed smFISH using a probe that targets the unique *circRERE2* BSJ to specifically detect the subcellular localization of *circRERE2* in rat neurons (Fig. 3D, E). Thereby, we observed *circRERE2* positive puncta in the cell soma and, occasionally in dendritic processes of hippocampal pyramidal neurons (see Fig. 3D, insets at higher magnification; Supplementary Fig. 5). Almost no puncta were observed when using a scrambled smFISH probe, demonstrating that we can faithfully detect *circRERE2* with our protocol (Fig. 3E). Significantly reduced *circRERE2* puncta density in *circRERE*sh neurons further confirmed the specificity of the *circRERE2* smFISH signal.

Given our RNA-seq results (Fig. 3B), we wondered whether there might be an additive effect on synaptogenesis by simultaneously knocking down the three most process-enriched *circRERE* 1–3 isoforms (referred to as *circRERE*sh from here on). The specificity and efficiency of the individual shRNA constructs targeting the different isoforms were validated by qPCR after transfection through nucleofection in rat cortical neurons (Supplementary Fig. 6). *circRERE2* knockdown alone resulted in a significant increase in synapse co-clusters compared to the ScramSh control (Fig. 3F, G; Supplementary Fig. 7A, B), thereby confirming our results from the siRNA screen (Fig. 2). Knockdown of all 3 *circRERE* isoforms resulted in a more robust increase in synapse co-cluster (Fig. 3G) and dendritic spine (Supplementary Fig. 7C) density compared to the *circRERE2* shRNA, indicating that *circRERE* isoforms 1 and 3 also contribute to the repression of excitatory synapse formation. However, synapse co-cluster density in circRERE sh neurons did not significantly differ from that in *circRERE2*sh neurons ($p = 0.8851$), suggesting that *circRERE2* is functionally the most relevant circular *RERE* isoform and that knockdown of the *circRERE2* isoform alone recapitulates most of the function of the entire *circRERE* isoform pool. The effect of *circRERE* KD was specific for synapse formation since siRNA-mediated knockdown of none of the circRNA isoforms 1–3 had a significant impact on dendritogenesis based on Sholl analysis (Supplementary Fig. 8A–E). In contrast, *linRERE* knockdown resulted in widespread neuronal death of transfected cells, preventing synaptic characterization, and suggesting that the observed increase in synapse co-clusters is not an artifact due to erroneous knockdown of the linear *RERE* mRNA.

Next, we explored whether increased synapse co-cluster density upon *circRERE* knockdown was accompanied by alterations in synaptic transmission using whole-cell patch-clamp electrophysiological recordings. *circRERE*sh transfected rat hippocampal neurons displayed a robust (49%) decrease in mEPSC frequency, but no significant change in mEPSC amplitude (Fig. 4A–E). This result implies that the vast majority of excess synaptic co-clusters formed upon *circRERE* knockdown might be functionally inactive ("silent"). We therefore decided to characterize excitatory synapses in these neurons in further detail. When plotting the mEPSC decay times, which depend on the subunit composition of AMPA-type glutamate receptors (AMPA-Rs)[36], we observed a significant decrease in *circRERE*sh compared to ScramSh transfected neurons (Fig. 4F, G), consistent with reduced GluA1 expression at *circRERE* knockdown synapses[37]. Notably, genetic deletion of GluA1 leads to the formation of non-functional, "silent" synapses[38].

To assess GluA1 expression in *circRERE*sh knockdown neurons more directly, we performed immunocytochemistry for surface-expressed GluA1 subunit (Fig. 4H, I). Transfection of *circRERE*sh resulted in a significant decrease in GluA1 expressed on the cell surface (Fig. 4I). Intriguingly, reduced surface GluA1 expression upon *circRERE* KD was accompanied by increased total GluA1 protein levels as detected by immunostaining of fixed neurons (Fig. 4J, K), as well as increased dendritic spine density (Supplementary Fig. 8F). This result suggests that *circRERE* inhibits GluA1 expression, but promotes its membrane localization. Together our results indicate a dual role of *circRERE* in excitatory synapse development by controlling both their formation and functional maturation, although in opposite directions.

## circRERE suppresses excitatory synapse formation by stabilizing miR-128-3p

circRNAs are able to affect gene expression by a variety of different mechanisms[39,40]. Therefore, we reasoned that we might obtain first insight into the mechanism of *circRERE* function by assessing the protein-coding transcriptome in neurons upon *circRERE* knockdown. Primary rat cortical neurons were electroporated with *circRERE2*sh (targeting the most abundant *circRERE* isoform) or two control constructs (empty vector (pSUPER), ScramSh), followed by total RNA extraction and polyA-RNA sequencing. Bioinformatic analysis of the RNA-seq data revealed a total of 869 genes differentially expressed (DEGs) between the *circRERE2*sh and control conditions (Fig. 5A). 374 were upregulated, and 495 were downregulated.

GO Term analysis of the DEGs was performed to delineate biological pathways regulated downstream of *circRERE* (Fig. 5B, Supplementary Fig. 9A, B). The top 15 GO terms in the category "cellular component" are mostly related to the synapse, with the top term being "glutamatergic synapse" (Fig. 5B). This was also supported by GSEA analysis with "GOCC_Synapse" as the stand-out gene set

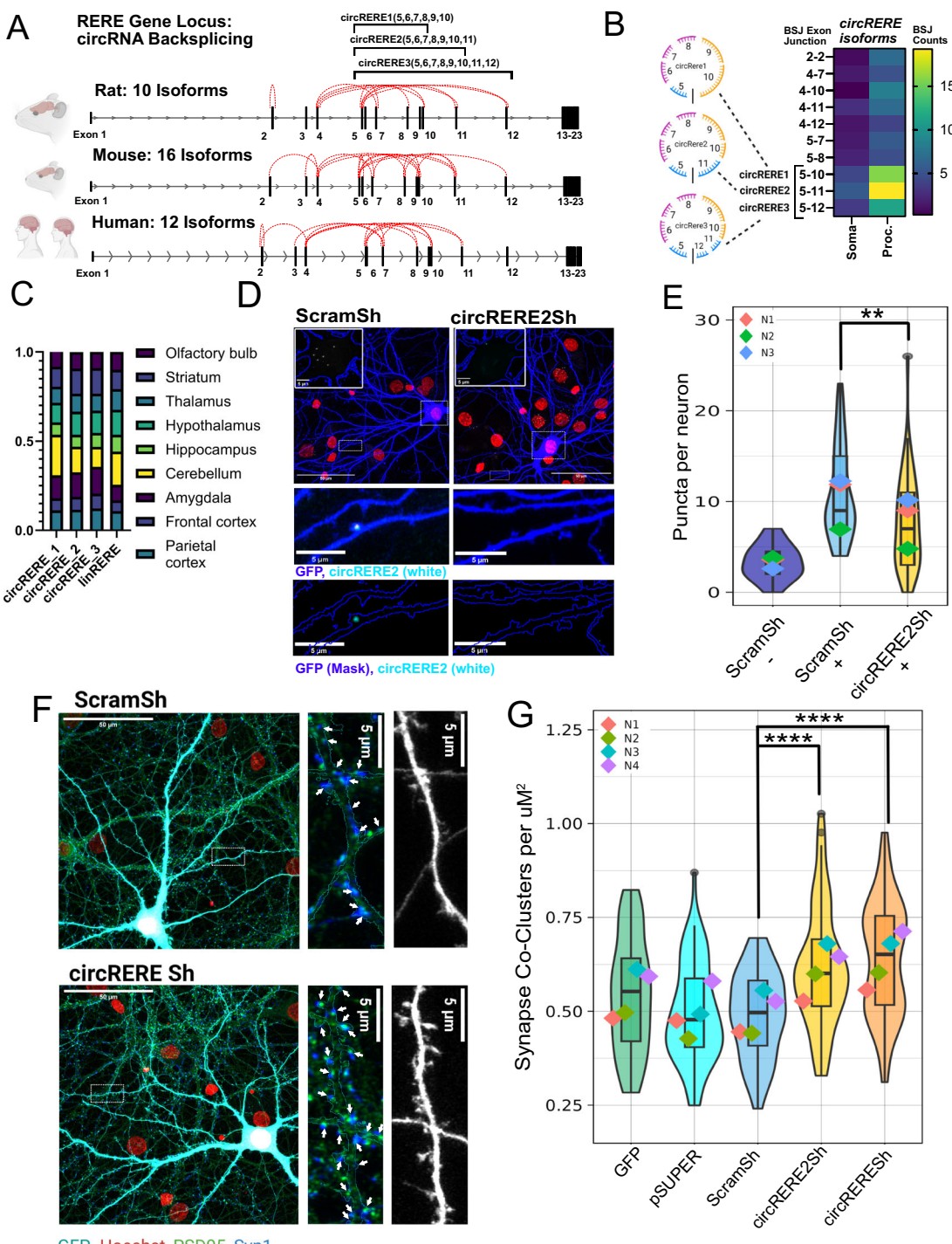

(Supplementary Fig. 9C). Together, these results are consistent with our results from the synaptic co-cluster analysis (Fig. 3F, G) and electrophysiology (Fig. 4A–G) and further support a role for *circRERE* in excitatory synapse development.

We were interested in how *circRERE* controls the expression of synaptic genes. Previously, several circRNAs have been shown to act as "miRNA sponges" by directly associating with specific miRNAs, e.g., *Cdr1-as*[7] and *circHipk3*[30]. We therefore used our previously developed bioinformatic webtool enrichmiR[41] to assess whether specific miRNA binding sites are overrepresented in DEGs over background. This would indicate altered activity of the corresponding miRNA in response to *circRERE* KD. EnrichmiR analysis revealed that particularly transcripts containing binding sites for the neuronal miRNA miR-128-3p

are significantly overrepresented in upregulated genes upon *circRERE2* kd (Fig. 5C). When plotting the expression fold-change for all genes as a function of the presence of miR-128-3p binding sites of different affinity (CD plot, Fig. 5D), we noticed that transcripts harboring particularly strong miR-128-3p binding sites (yellow curve, see figure legend for further explanation of scores) were on average more strongly upregulated in the *circRERE* kd neurons compared to those containing either weak or no sites (red and black curves). Thus, *circRERE* kd presumably leads to a lower miR-128-3p repressive activity, which is consistent with a stabilizing effect on miR-128-3p that is mediated by *circRERE*.

As *circRERE2* kd presumably results in a reduction in miR-128-3p levels/activity, we focused on upregulated genes in the synapse GO

**Fig. 3 | circRERE isoforms are abundant in neuronal processes and negatively regulate excitatory synapse co-clusters. A** Illustration of the conserved *RERE* gene locus and circRNA back-splicing between different exons, with emphasis on the major process-enriched isoforms *circRERE* 1, 2, and 3. **B** *circRERE* isoform dendritic and soma expression heatmap comparing the unique BSJ counts, showing the highest expression in the process compartment for *circRERE2*, followed by *circRERE* 1 and 3. **C** *circRERE* isoforms and *linRERE* relative expression in adult rat brains (*n* = 1) based on qPCR (normalized to Gapdh). **D** Representative images of *circRERE2* smFISH in ScramSh and *circRERE*Sh transfected rat hippocampal neurons (DIV16) using a specific BSJ probe. Blue: GFP; turquoise: *circRERE2* FISH; red: Hoechst. Upper panel: overview images and boxed insets showing somatic *circRERE2* positive puncta. Lower panel: magnification of boxed insets showing *circRERE2* positive puncta inside dendrites. **E** Quantification of *circRERE2* puncta from neurons transfected as in (**D**). − Condition utilizes a control probe. + Conditions utilize a *circRERE2* BSJ targeting probe. *N* = 3 independent biological replicates, 12–17 cells per condition, 5 Cells per condition for control probe condition. Violin plots with embedded boxplots, including the median, interquartile range, and whiskers from minimum to maximum. GLMM statistical modeling (two-sided). Emmeans: ScramSh− = 3.33 ± 1.77. ScramSh+ = 10.44 ± 1.47. circRERE2Sh+ = 7.90 ± 1.50.

Estimatediff: ScramSh− vs. ScramSh+: = −7.11 ± 1.274, *t*-ratio = −5.583, *p* < 0.0001. ScramSh− vs. circREREsh+: estimatediff = −4.57 ± 1.304, *t*-ratio = −3.504, *p* = 0.0019. ScramSh+ vs. circREREsh+: estimatediff = 2.54 ± 0.854, *t*-ratio = 2.979, *p* = 0.0097. *P* < 0.01 = ** *P* =/< 0.0001 = ****. **F** Representative images from neurons transfected with ScramSh or *circRERE*sh and stained for Syn1 (blue)/PSD-95 (green). Turquoise: GFP; red: Hoechst. Left panel: overview. Middle and right panel: Boxed insets at higher magnification depicting synapse co-clusters (arrows, middle) or dendritic spines (right) within the GFP mask. **G** Quantification of PSD-95/Synapsin co-cluster density in rat hippocampal neurons (DIV16) transfected with the indicated shRNAs or control plasmids. 15 Cells per condition per replicate, *N* = 4 independent biological replicates. Violin plots with embedded boxplots, including the median, interquartile range, and whiskers from minimum to maximum. GLMM statistical modeling (two-sided). Emmeans: GFP = −0.933 ± 0.0858, pSUPER = −0.0858 ± 5.33, ScramSh = −1.089 ± 0.0855, *circRERE2*Sh = −0.753 ± 0.0855, *circRERE*sh = −0.690 ± 0.0855. Contrasts: ScramSh/*circRERE2*Sh: estimatediff = −0.3359 ± 0.0672, *t*-ratio = −4.998, *p* = 0.0001. ScramSh/*circRERE*sh: estimatediff = −0.3984 ± 0.0672, *t*-ratio = −5.928, *p* = 0.0001. *P* </= 0.0001 ****. Created in BioRender. https://BioRender.com/d94h458. Source data are provided as a Source Data file.

---

term (*n* = 20; Fig. 5E). Among those, five (*Adam10, Fxr2, Gria1, Lin7C and Gabrg2*) contain canonical, high-affinity miR-128-3p binding sites in their 3′ UTR (Fig. 5F). Upregulation of *Gria1* mRNA is consistent with our results from immunostaining, whereby we observed increased total levels of the *Gria1*-encoded GluA1 protein in *circRERE* KD neurons (Fig. 4J, K). Thus, upregulation of this group of synaptic genes upon perturbation of a protective *circRERE*/miR-128-3p interaction might underlie the observed synaptic phenotypes.

To investigate the impact of *circRERE* knockdown on the expression of mature miRNAs, we subjected the same samples used for polyA-RNA-seq to small RNA-seq. Differential expression analysis revealed that the expression of miR-128-3p (derived from the two precursors *rno-miR-128-1* and *rno-miR-128-2*) (Fig. 6A) was indeed highly significantly reduced. Based on polyA-RNA-seq, transcripts originating from both of the miR-128-3p host genes (*Arpp21, R3hdm1*) are not significantly downregulated in *circRERE* knockdown neurons, arguing against an effect on miR-128-3p transcription or processing. In fact, the host of 128-2-3p, *Arpp21*, is itself regulated by 128-3p via a negative feedback process[42]. Accordingly, upregulation of *Arpp21* (FDR < 0.0674) can be observed upon *circRERE* knockdown. *R3hdm1*, the host of miR-128-1-3p, is unaffected (Supplementary Fig. 9D). Together, our results are consistent with a role for *circRERE* in the stabilization of the mature miR-128-3p.

Next, we addressed more directly whether the observed reduction in miR-128-3p levels upon *circRERE* knockdown translated into reduced miR-128-3p repressive activity. Therefore, we performed luciferase assays with a 2× perfect miRNA binding site reporter in the context of shRNAs targeting either the predominant *circRERE* isoforms 1–3 or linear *RERE* mRNA (Fig. 6B, C). Combined knockdown of all *circRERE* isoforms with *circRERE*sh significantly derepressed the miR-128-3p reporter (Fig. 6C), suggesting that *circRERE*-mediated stabilization of miR-128-3p is associated with increased miR-128-3p activity. Knocking down the individual isoforms *circRERE 1* and *circRERE2*, but not *circRERE3*, was sufficient to significantly increase miR-128-3p luciferase reporter activity (Supplementary Fig. 10A). This result correlates well with our previous results regarding the function of specific *circRERE* isoforms in synapse formation (Fig. 2C).

To assess the effects of *circRERE* KD on neuronal miR-128-3p expression at the subcellular level, we performed smFISH with a probe specific for mature miR-128-3p (Fig. 6D–F). *circRERE*sh resulted in an average 47% decrease in miR-128-3p positive puncta in hippocampal pyramidal neurons, highly consistent with our results from small RNA-seq (Fig. 6A). Overall, the magnitude of reduction was similar between the somatic and dendritic compartment, although the vast majority of

puncta (~90%) were detected within the soma. The specificity of the miR-128-3p signal was confirmed by performing smFISH with a control probe ("ScramSh - condition"), which resulted in a low background signal (Fig. 6E, F). Together, our smFISH results show that *circRERE2* and miR-128-3p engage in a protective interaction in both the somatic and dendritic compartments of rat hippocampal neurons.

Given the pronounced effect of *circRERE* knockdown on miR-128-3p expression, we asked whether restoring miR-128-3p levels was sufficient to rescue the synapse phenotype caused by *circRERE* knockdown. Toward this end, we also utilized a previously published plasmid for miR-128-3p overexpression (dsRED-miR-128-3p, from here on called miR-128OE)[42]. Using the 2× miR-128-3p PBS luciferase assay, we validated the expression of functional miR-128-3p by miR-128OE (Supplementary Fig. 10B). Moreover, when we co-express miR-128-3p OE together with *circRERE*sh, excessive synapse co-cluster density caused by *circRERE* knockdown alone returned to baseline levels, demonstrating efficient rescue (Fig. 6G, H; Supplementary Fig. 10C–E). Interestingly, miR-128OE alone did not further decrease synapse density, possibly due to its high expression already at basal levels.

Together, our results strongly support the hypothesis that the synaptic phenotype caused by *circRERE* knockdown is mediated by a decrease in miR-128-3p levels, either occurring at the somatic or dendritic level.

## circRERE function in excitatory synapse formation is dependent on the presence of intact miR-128-3p binding sites

We wondered whether *circRERE* stabilizes miR-128-3p levels via a direct interaction, i.e., protecting it from destruction via endogenous decay pathways. Using ScanmiR, a bioinformatic tool developed in our lab that identifies predicted miRNA binding sites within transcripts[43], we detected two putative high-affinity miR-128-3p binding sites within exon 9 of *circRERE* (Fig. 7A). Importantly, these sites are located in exon 9 of the *RERE* gene which is present in all of the highly expressed *circRERE* isoforms 1–3. In addition, an AGO HITS-CLIP peak is present on *RERE* exon 9 between these sites[44], further supporting a direct interaction between *circRERE* and miRNAs.

To investigate the functional relevance of these two miR-128-3p binding sites, we generated *circRERE* overexpression constructs utilizing the Wilusz lab's Zkscan MCSOE vector[45], containing *RERE* exons 5–12 (corresponding to *circRERE3* with a different BSJ sequence), either containing wild-type (referred to as *circRERE* WT) or mutant (referred to as *circRERE* 128-MUT) versions of the 7mer and 8mer miR-128-3p binding sites detected on exon 9 (Fig. 7A). The efficacy and specificity

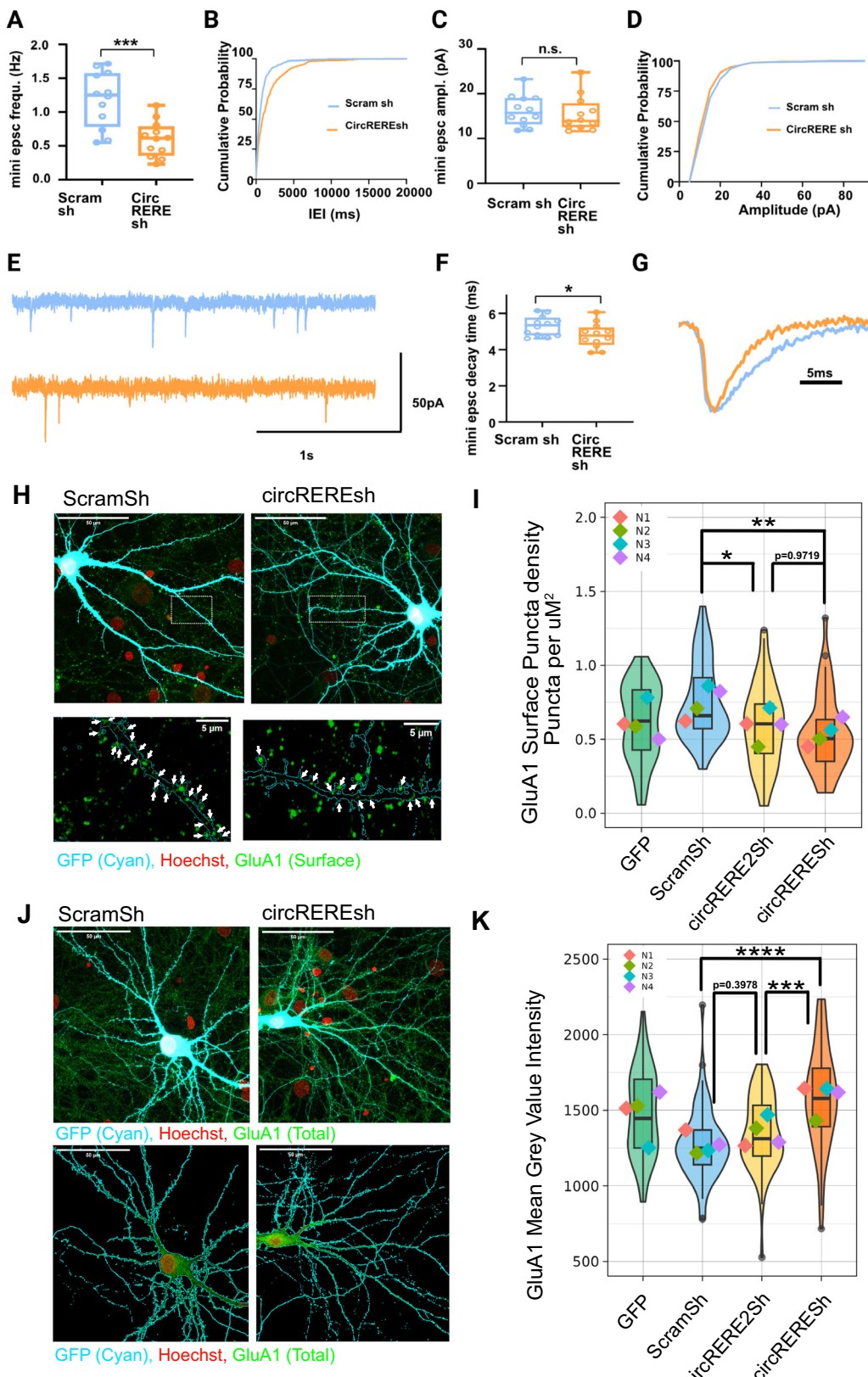

of *circRERE* WT and *circRERE* 128-MUT construct overexpression was validated following electroporation of primary cortical neurons by RT-qPCR using primers detecting specifically *circRERE3*, which is expressed at very low levels endogenously (Supplementary Fig. 11A, B). To assess a potential expression of linear *RERE* transcripts from our Zkscan MCSOE vector, which could confound our results, we

determined the ratio of overexpressed *circRERE3* and the total amount of circular and linear *RERE* transcripts (both endogenous and over-expressed) in neurons transfected with *circRERE* WT or 128-MUT (Supplementary Fig. 11C–G). This revealed that a substantial fraction (about 80–90%) of all *RERE* transcripts represent overexpressed *circRERE3*. More importantly, this fraction did not change upon RNAse R

**Fig. 4 | *circRERE* knockdown in hippocampal neurons leads to reduced excitatory synaptic transmission and GluA1 surface expression.** Electrophysiological characterization of *circRERE* kd neurons. **A, B** mEPSC frequency. $p = 0.0004$, ScramSh = $1.185 \pm 0.1411$, *CircRERE*sh = $0.5992 \pm 0.1411$, $t = 4.154$, df = 22. 95% confidence interval −0.8788 to −0.2935. (two-sided unpaired *t*-test). **C, D** mEPSC amplitudes $p = 0.7452$. ScramSh = $1.625 \pm 0.1516$, *CircRERE*sh = 15.44 $\pm 0.1516$, $t = 0.5322$, df = 22. 95% CI −3.950 to 2.337 (two-sided unpaired *t*-test). **E** Representative traces illustrating mEPSC frequency decreases with *circRERE*sh compared to ScramSh. **F, G** mEPSC decay time in *circRERE* shRNA (orange bars) or shRNA control (blue bars), $p = 0.0485$. ScramSh = $5.297 \pm 0.2493$, *CircRERE*sh = $4.776 \pm 0.2493$, $t = 2.089$, df = 22. 95% CI, −1.038 to −0.003845 (two-sided unpaired *t*-test). **A, C, F** $N = 12$, whole-cell patch-clamp recordings of primary hippocampal neurons, boxplots including the median, interquartile range, and whiskers from minimum to maximum. $P < 0.05$ *, $P < 0.001$ ***. **H** Representative images from rat hippocampal neurons (DIV16) transfected with ScramSh or *circRERE*sh and immunostained for surface GluA1 (green). Turquoise: GFP; Red: Hoechst. Upper panel: Overview image. Lower panel: Boxed inset at higher magnification showing GluA1 puncta (green, arrows) located at the dendritic surface. **I** Quantification of GluA1 surface puncta density in neurons transfected with indicated *circRERE* shRNA constructs. $N = 4$ independent biological replicates, 10 cells per condition per experiment. Violin plots with embedded boxplots, including the median, interquartile range, and whiskers from minimum to maximum. GLMM statistical modeling (two-sided). Emmeans: GFP = −0.881 ± 0.132, ScramSh = −0.491 ± 0.132, *circRERE2*Sh = −0.983 ± 0.132, *circRERE*sh = −1.059 ± 0.132. Contrasts: ScramSh/*circRERE2*Sh = 0.493 ± 0.174, *t*-ratio = 2.838, $p = 0.0263$. ScramSh/*circRERE*sh = 0.569 ± 0.174, *t*-ratio = 3.275, $p = 0.0071$. *circRERE2*Sh/*circRERE*sh = 0.076 ± 0.174, *t*-ratio = 0.438, $p = 0.9719$. GLMM statistical modeling. $P < 0.05$ = *, $P < 0.01$ = **. **J** Representative images from rat hippocampal neurons (DIV16) transfected with ScramSh or *circRERE*Sh and immunostained for total GluA1 (green). Magenta: GFP. **K** Quantification of total GluA1 mean gray value intensity in cells transfected with indicated *circRERE* shRNA constructs. GLMM statistical modeling (two-sided) performed post $\log_2$ transformation of raw-values. Emmeans: GFP = 10.5 ± 0. 0378, ScramSh = 10.3 ± 0.0378, *circRERE2*Sh = 10.4 ± 0.0378, *circRERE*sh = 10.6 ± 0.0378. Contrasts: ScramSh/*circRERE2*Sh = −0.0839 ± 0.0534, *t*-ratio = −1.570, $p = 0.3978$. ScramSh/*circRERE*Sh = −0.3040 ± 0.0534, *t*-ratio = −5.692, $p < .0001$. *circRERE2*Sh/*circRERE*Sh = −0.2202 ± 0.0534, *t*-ratio = −4.122, $p < 0.0003$. $p < .0001$ = ***. $P < 0.0001$ = ****. Source data are provided as a Source Data file.

treatment, demonstrating that the contribution of linear transcripts to the entire *RERE* transcript pool is negligible.

Next, we explored the effect of *circRERE* overexpression on miR-128-3p expression by miRNA qPCR (Fig. 7B). *circRERE* WT led to a significant increase in miR-128-3p levels compared to the MCSOE control, demonstrating that *circRERE* is not only necessary but also sufficient to stabilize miR-128-3p. Importantly, no significant difference was observed between MCSOE control and *circRERE* 128-MUT expression. Together, this supports the hypothesis of a protective interaction between *circRERE* and miR-128-3p in a manner dependent upon the presence of the predicted two miR-128-3p miRNA binding sites. Having shown the differential effects exerted by the *circRERE*-WT and 128-MUT constructs on miR-128-3p expression, we further tested their ability to rescue excessive synapse co-cluster density caused by *circRERE* knockdown. Importantly, both constructs are resistant to the *circRERE* shRNA due to their unique BSJ. We expected that an efficient rescue should only be achieved with the *circRERE* WT-overexpression since the presence of miR-128-3p binding sites within *circRERE* is necessary for miR-128-3p stabilization (Fig. 7B). In agreement with this assumption, we observed a rescue of synapse co-cluster density with *circRERE* WT, but not *circRERE* 128-MUT, when co-expressed with *circRERE*sh (Fig. 7C, D; Supplementary Fig. 12).

In conclusion, these results tie the presence of miR-128-3p binding sites within *circRERE* to its effects on excitatory synapse formation. This strongly supports a model whereby *circRERE* protects miR-128-3p from degradation via direct interaction at baseline, thereby stabilizing neuronal mir-128-3p levels and restricting synapse formation via the downregulation of critical miR-128-3p synaptic target mRNAs at the peak of excitatory synaptogenesis in vitro (Fig. 8).

## Discussion

In this study, we performed a systematic characterization of neuronal circRNAs during excitatory synaptogenesis. By using a compartmentalized primary rat hippocampal neuron model, we identified several circRNAs as inhibitors of excitatory synapse formation and further characterized the molecular mechanism of one of these candidates, *circRERE*. Thus, our study demonstrates a significant contribution of circRNAs to the regulation of mammalian synaptogenesis and provides insight into their mode of action, specifically regarding their regulation of miRNA activity. This will form the basis for follow-up studies on their (patho)physiological relevance using in vivo models.

### circRNAs are enriched in the synapto-dendritic compartment

For the identification of process-enriched circRNAs, we leveraged a previously described primary rat hippocampal neuron model which allows the physical separation of processes (axons, dendrites) from somata. Since this system also contains glial processes[25], circRNAs detected by our RNA-seq approach could, in principle, also originate from glial cells. However, 113/234 processed enriched circRNAs were also previously detected in a mouse synaptoneurosome preparation[5], including 8 *circRERE* isoforms. This includes 31/32 screened circRNAs in our study (with the exception of *circAnkrd12*). Moreover, we (Fig. 1E) and others[6] could validate the dendritic localization of several candidates by smFISH. Therefore, the vast majority of the process-enriched circRNAs are likely of neuronal origin.

Our work found that a large fraction (22%) of neuronal circRNAs displays a high enrichment in the process compartment when compared to either the overall transcript population (RNA-seq) (Fig. 1B), their linear counterparts (Fig. 1C), or individual "house-keeping" genes (Gapdh; Fig. 1D). This apparent overrepresentation of circRNAs in processes implies the existence of specific circRNA transport mechanisms which remain to be determined and might, for example, involve post-transcriptional m6A modifications[46] or the interaction with RNA-binding proteins (RBP)[47]. Alternatively, since circRNAs are inherently more stable than other RNA types, they could gradually accumulate over time in the synapto-dendritic compartment. We favor the first model since it is more consistent with the observation that a small subset of neuronal circRNAs (e.g., *circSnap25, Capns1, Ubr5, Nckap1*) is actively depleted from processes, which would argue against passive diffusion. What might be the functional relevance of this compartment-specific circRNA enrichment? One can speculate that it might skew the stoichiometric ratio of circRNAs and their interaction partners (miRNA or RBP) in favor of circRNAs, thereby allowing for a more efficient functional interaction, i.e., in the context of local protein synthesis regulation at synapses.

### circRNAs act as negative regulators of excitatory synapse formation

This study systematically screened for functional circRNAs in post-mitotic neurons. While systematic functional screens for circRNAs are generally scarce, RNAi-based[48] and CRISPR-Cas13-based screens[49] have been primarily undertaken in cancer cells. Interestingly, our main hit, *circRERE*, has been identified as a cell-essential circRNA that regulates ferroptosis in cancer cell lines[48]. Intriguingly, our screen results show that circRNAs have an overall repressive impact on synapse formation, possibly preventing the premature and/or excessive formation of synapses during neuronal development. When interrogating the functional implications with electrophysiological recordings for the candidate *circRERE*, we found that increased synapse co-cluster density upon *circRERE* knockdown was associated with reduced mEPSC

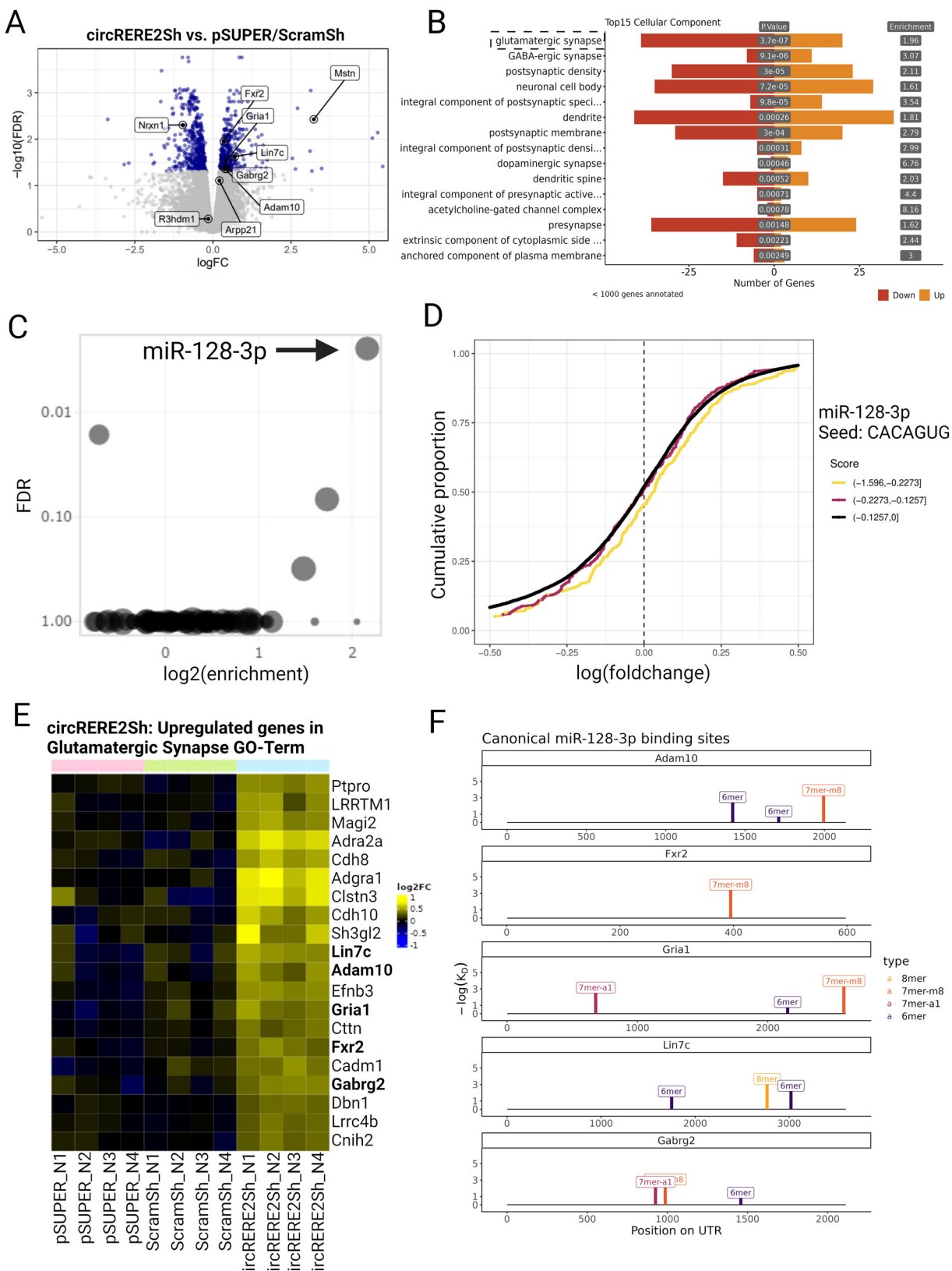

frequency. This finding is most consistent with the idea that these newly formed synaptic co-clusters are non-functional due to deficits in postsynaptic GluA1 expression ("silent synapses"). However, at this point, we cannot exclude additional or sole deficits in presynaptic release. A more in-depth electrophysiological characterization is required to distinguish between these possibilities. In any case, this

suggests a dual function of *circRERE*, working both as a repressor of synapse formation but also as a promoter of synapse maturation. We speculate that such a function could be important to ensure the homeostasis of neural circuit activity during development.

Besides the circRNAs described here, only a few additional circRNAs have previously been shown to be involved in the regulation of

**Fig. 5 | *circRERE2* knockdown results in a preferential upregulation of miR-128-3p targets involved in excitatory synapse function. A** Volcano plot showing significant differentially expressed genes (DEGs; in blue) between *circRERE2*sh and control pSUPER/ScramSh electroporated rat cortical neurons (DIV5). Statistical analysis was performed with Genewise Negative Binomial Generalized Linear Models, FDR cutoff < 0.05, no logFC cutoff. **B** Bar plot showing the number of genes in each significant GO term from cellular compartment ontology; orange bars = upregulated genes, red bars = downregulated genes (one-sided). **C** EnrichMiR analysis of miRNA binding sites enriched in DEGs shown in (**A**). **D** Cumulative

distribution (CD)-plot of log fold-changes from DEGs shown in (**A**), containing either low (black line; affinity score 0 to −0.1257), medium (red line; affinity score −0.1257 to −0.22723) or high (yellow line; affinity score −0.2273 to −1.596) affinity miR-128-3p binding sites. The right shift of the curves indicates, on average, higher expression of miR-128-3p site containing transcripts in *circRERE* knockdown neurons. **E** Heatmap of selected DEGs from (**A**) belonging to the glutamatergic synapse GO term (FDR < 0.05, upregulated only). **F** Illustration of miR-128-3p binding sites (separated by site type) within the 3′ UTR of the indicated upregulated DEGs. Source data are provided as a Source Data file.

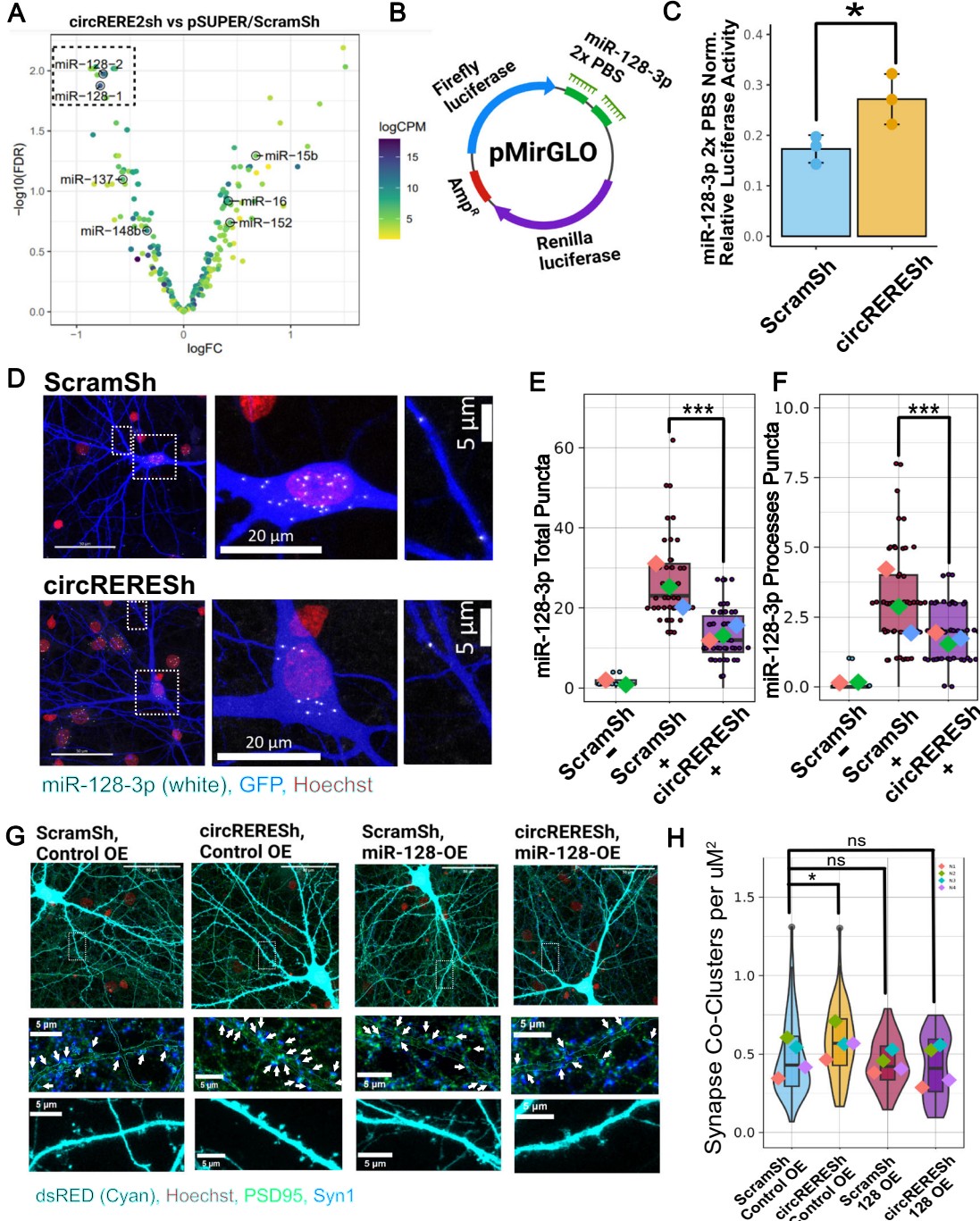

**Fig. 6 | *circRERE* knockdown results in reduced miR-128-3p expression levels and activity. A** Volcano plot showing differentially expressed miRNAs between *circRERE2*sh/control shRNA-transfected neurons. Statistics performed with Gene-wise Negative Binomial Generalized Linear Models. Highly downregulated miR-128-1 and -2 encoding for miR-128-3p (identical sequences) are highlighted. **B** Illustration of 2× Perfect binding site Dual-luciferase construct sensitive to miR-128-3p activity (miR-128-3p 2× PBS). **C** Relative luciferase activity in neurons transfected with miR-128-3p 2× PBS and indicated shRNA constructs, $N = 3$. Bar graphs indicate mean +/- s.d. Two-sample $t$-test, Emmeans: ScramSh = 0.173, *circcRERE*sh-0.2716667, $t = -2.9948$, $P = 0.04015$ $P < 0.05 = *$. **D** Representative images from miR-128-3p smFISH in ScramSh (upper-row) or *circRERE*sh (lower-row) transfected rat hippocampal neurons. Blue: GFP; white: miR-128-3p smFISH; red: DAPI. Left panel: overview images. Middle and right panels: Boxed insets at higher magnification show somatic (middle) or dendritic (arrow, right) miR-128-3p positive puncta. **E** Quantification of miR-128-3p puncta in somata (**E**) or processes (**F**) of neurons transfected as in (**D**). GLMM (two-sided). **E, F** $N = 3$ Biological Replicates, 15–20 cells per condition per replicate. Exception: $N = 2$, 6–7 cells per condition per replicate for ScramSh− control probe condition. Boxplots include median, interquartile range, and whiskers from minimum to maximum. Total Puncta: Emmeans: ScramSh− = 1.15 ± 2.53, ScramSh+ = 25.90 ± 1.36, *circRERE*sh+ = 13.37 ± 1.36. Contrasts: ScramSh−/ScramSh+ = −24.7 ± 2.72, $t = −9.101$, $p < 0.0001$. ScramSh−/*circRERE*sh+ = −12.2 ± 2.72, $t = −4.493$, $p = 0.0001$, ScramSh+/*circRERE*sh+ = 12.5 ± 1.7, $t = 7.354$, $p = 0.0001$. $P = 0.001$ ***, $P < 0.0001$ ****. Datapoints represent the number of puncta detected per neuron. Processes Puncta: Emmeans: ScramSh − = −0.0964 ± 0.516, ScramSh+ = 3.0576 ± 0.576, *circRERE*sh+ = 1.7107 ± 0.376. Contrasts: ScramSh−/ScramSh+ = −1.81 ± 0.452, $t = −6.976$, $p = 0.0001$. ScramSh −/*circRERE*sh+ = 1.35 ± 0.285, $t = −3.997$, $p = 0.0003$, ScramSh+/*circRERE*sh + = 12.5 ± 1.7, $t = 4.724$, $p = 0.0001$. **G** Representative images from rat hippocampal neurons (DIV16) transfected with indicated shRNAs and overexpression (OE) plasmids. Turquoise: GFP; green: PSD-95; blue: synapsin-1; red: Hoechst. Upper panel: overview image. Middle/lower panels: boxed insets at higher magnification showing synapse co-clusters (arrows, middle) or dendritic segments with spines (lower) within the GFP mask. **H** Quantification of PSD-95/Synapsin co-clusters in $N = 4$ independent experiments, 12–16 Cells per experiment and condition. Violin plots with embedded boxplots, including median, interquartile range, and whiskers from min-max. GLMM statistical modeling (two-sided). Emmeans: ScramSh,ControlOE = −1.241 ± 0.18, *circRERE*sh,ControlOE = −0.927 ± 0.182, ScramSh,128OE = −1.261 ± 0.181, *circRERE*sh,128OE = −1.439 ± 0.180. Contrasts: *circRERE*sh, ControlOE/ScramSh, ControlOE = 0.314 ± 0.121, $t = 2.586$, $p = 0.0287$. ScramSh,128OE/ScramSh, ControlOE = −0.20 ± 0.12, $t = −0.167$, $p = 0.9891$, *circRERE*sh,128OE/ScramSh,ControlOE = −0.198 ± 0.119, $t = −1.666$. $p = 0.2350$. $P < 0.05$ *. Created in BioRender. https://BioRender.com/j06t098. Source data provided as Source Data file.

---

synapse development. *Cdr1-as* is arguably the most characterized circcRNA and has been shown to regulate the activity of miR-7 in an activity-dependent manner[10]. *Cdr1-as* knockout in mice leads to dysfunction in excitatory synaptic transmission with an increase in spontaneous excitatory postsynaptic currents[8] congruent with our screen, where we picked up *Cdr1-as* as the top regulator of dendritic spine size and number (Supplementary Fig. 3B, C). Further examples include *CircGria1*, whose knockdown increases synaptogenesis in the macaque brain, presumably via upregulation of linear *Gria1* mRNA from its host gene[13]. On the other hand, the dendritically enriched circHomer1 was shown to regulate synaptic genes by interacting with the RBP HuD[15], although the impact on synaptic transmission was not directly addressed in this study. Very recently, *circSatb1* and *CircDlc1(2)*, both of which were not part of our screen, were shown to regulate dendritic spine morphology and glutamatergic transmission in culture, respectively[12].

### circRERE function in synaptogenesis requires miR-128-3p

We provide multiple lines of evidence that the *circRERE* function critically relies on the stabilization of the neuronal miRNA miR-128-3p (Fig. 8). miR-128-3p is abundantly expressed in rodent neurons[50] and has been repeatedly involved in the regulation of synapse development and function. In agreement with our observation of reduced miR-128-3p expression upon *circRERE* knockdown, miR-128-3p deficiency in D1 dopaminergic neurons results in increased dendritic spine number[51]. However, miR-128-3p deficiency was further shown to lead to increased neuronal excitability in multiple studies, which culminates in an increased susceptibility for epileptic seizures[42,51,52]. This would suggest that the effects of *circRERE* on synapse formation and function are uncoupled and display differential dependency with regards to miR-128-3p. To address this, additional experiments on the impact of miR-128-3p on synapse physiology in the context of *circRERE* knockdown are required.

RNA sequencing revealed interesting miR-128-3p targets which could mediate the positive effects of *circRERE* knockdown on excitatory synapse formation (Fig. 5). For example, FXR2 was shown to promote the activity-dependent local translation of PSD-95, a central component of the postsynaptic density[53]. The AMPA-type glutamate receptor subunit GluA1, which is encoded by the *Gria1* gene, is a critical determinant of dendritic spine structure and excitatory synapse function[54]. Interestingly, despite increased *Gria1* mRNA (Fig. 5E) and GluA1 protein (Fig. 4K) levels, we observed less GluA1 surface expression upon *circRERE* knockdown (Fig. 4H, I). This is in agreement with our results from electrophysiological recordings (Fig. 4A–G) and provides further support for the existence of different pathways controlled by *circRERE* during synapse formation and maturation. This might involve the matrix metalloprotease Adam10, which has been shown to control the localization of GluA1 to synapses and to remodel dendritic spines through the cleavage of adhesion molecules, such as N-Cadherin[55]. It will be interesting to test which of these miR-128-3p targets is causally involved in *circRERE*-mediated effects on either synapse formation or functional maturation.

### circRERE acts via miR-128-3p stabilization

The reported molecular mechanisms of circcRNA action are diverse, ranging from translational regulation to protein binding and miRNA sponging, with the latter being arguably the most widely described. For most cases, a sponging interaction between circRNAs and miRNAs has been shown to reduce the activity of the corresponding miRNAs on their targets, e.g., *circHipk3* and *circSry*[7,30]. However, the consequence of these interactions is not as straightforward as first envisioned. For example, *Cdr1-as* was originally conceived as a classical miRNA "sponge"[7], but further work rather suggests a protective interaction between *Cdr1-as* and miR-7. According to this, *Cdr1-as* protects miR-7 from TDMD-mediated degradation by the lncRNA *Cyrano*[8,9]. A similar protective interaction has also been recently described for *circCSNK1G3*, which harbors strong TDMD-like sites for Mir-181b/d. However, despite increased miR-181 levels, *circCSNK1G3* expression resulted in an overall decrease in target gene repression[56].

In the case of *circRERE*, our data currently supports a stabilizing interaction between *circRERE* and miR-128-3p, similar to what was recently observed for *Cdr1-as*/miR-7[10] and *circDlc1(2)*/miR-130b-5p[16]. Interestingly, *circRERE* knockdown affected miR-128-3p levels similarly in the dendritic and somatic compartments (Fig. 6E, F), suggesting that *circRERE* stabilizes miR-128-3p in both compartments. The fact that the vast majority of *circRERE*/miR-128-3p puncta are detectable in the somatic compartment does not per se rule out an important function of the *circRERE*/miR-128-3p interaction in dendrites or even at synapses. For example, *circRERE* might control the local availability of miR-128-3p at specific synapses in response to activity, similar to what has been recently described for *circDlc1(2)* and its interaction partner miR-130b-5p[16]. Moreover, the miR-128-3p target mRNA pool might vary between the somatic and dendritic compartments, with important

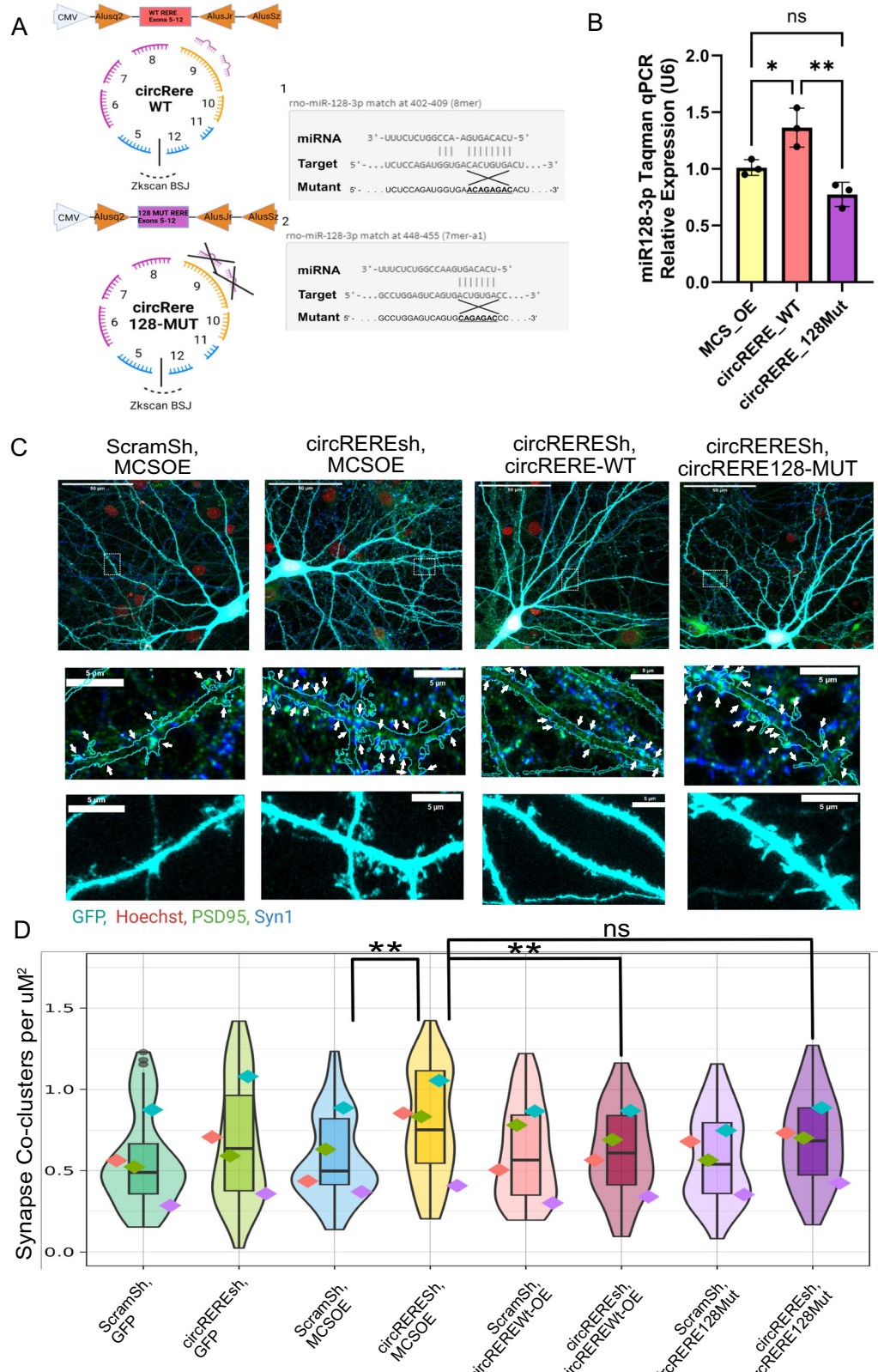

GFP, Hoechst, PSD95, Syn1

implications for the functional consequences of miR-128-3p regulation. Future experiments that more directly address the local translation of miR-128-3p targets in the soma and dendrites, preferentially in the context of neuronal activity changes, should help to resolve this issue. How miR-128-3p is degraded upon loss of the protective interaction with *circRERE* is currently unknown. In contrast to miR-7, miR-

128-3p is insensitive to knockout of the TDMD factor Zswim8, arguing that Zwim8-independent miRNA degradation pathways might be involved[57]. From a broader perspective, it will be interesting to know how widespread circRNA-mediated miRNA stabilization is, i.e., by studying additional synapse-regulating circRNAs and their predicted miRNA associations.

**Fig. 7** | **circRERE function in excitatory synapse formation is dependent on the presence of intact miR-128-3p binding sites. A** Schematic of the *circRERE* Wt and 128-Mut overexpression (OE) constructs. The exact sequences of the two miR-128-3p binding sites within exon 9 of *circRERE* are shown in their wild-type and mutant configuration on the right. **B** miR-128-3p expression levels determined by TaqMan miRNA qPCR in primary cortical neurons (DIV5) electroporated with MCSOE, *circRERE* WT, or 128-Mut OE constructs. U6 snRNA was used for normalization. $N = 3$ biological replicates. Bar graphs indicate mean +/- s.d. One-Way ANOVA, Tukey's multiple comparisons test, MCS_OE/*circRERE*_WT: Meandiff = $-0.3529 \pm 0.101$, $q = 4.939$, $p = 0.03$. MCS_OE/*circRERE*_128Mut: Meandiff = $0.2363 \pm 0.101$, $q = 3.308$, $p = 0.1254$. *circRERE*_WT/*circRERE*_128Mut: Meandiff = $0.5892 \pm 0.101$, $q = 8.247$, $p = 0.0027$. $P < 0.05$ *, $P < 0.01$ **. **C** Representative images of PSD-95/Synapsin co-clusters in rat hippocampal neurons (DIV16) transfected with the indicated shRNAs and circRNA expression plasmids. MCS: multiple cloning site. turquoise: GFP;

green: PSD-95; blue: Synapsin-1; red. Hoechst. Upper panel. Overview image. Middle and lower panel: Boxed insets at higher magnification depicting either synapse co-clusters (arrows, middle) or dendritic segments with spines (lower) within the GFP mask. **D** Quantification of synapse co-cluster density in neurons transfected with the indicated constructs as in (**C**). $N = 4$ biological replicates, 15 cells per condition. Violin plots with embedded boxplots, including the median, inter-quartile range, and whiskers from minimum to maximum. Pairwise comparisons from emmeans/GLMM (two-sided). Contrasts: MCSOE,*circRERE*sh/MCSOE,-ScramSh = Est:$0.20586052 \pm 0.05978018$ $t = 3.444$, $p = 0.0024$, FDR = 0.0170461. *circRERE* Wt-OE,*circRERE*sh/MCSOE,*circRERE*sh: est = $-0.17058049 \pm 0.05978018$, $t = -2.853$ $p = 0.0095$, FDR = 0.0333005. MCSOE,*circRERE*Sh/*circRERE*128Mut-OE,circREREsh: est = $0.10081020 \pm 0.05978018$, $t = 1.686$, $p = 0.1065$, FDR = 0.1531498. $P < 0.01$ **; ns not significant. Created in BioRender. https://BioRender.com/x08g762. Source data are provided as a Source Data file.

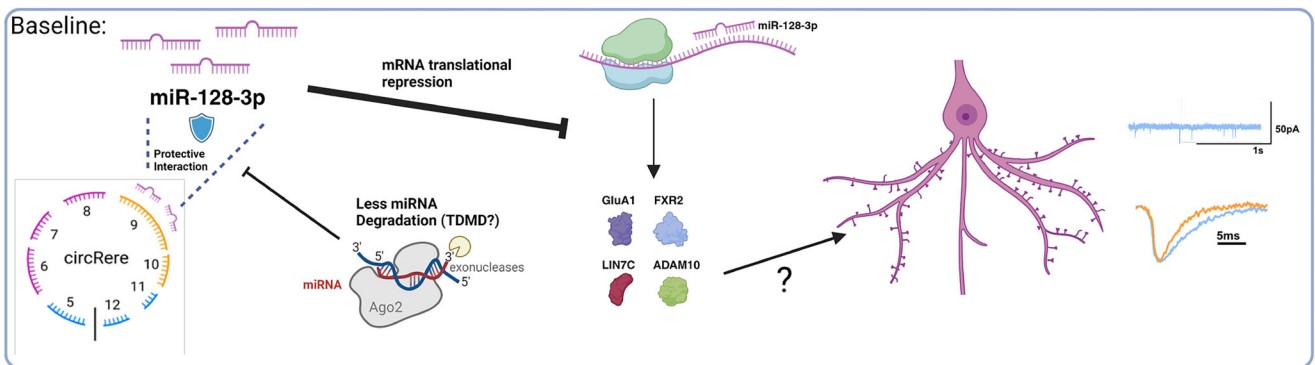

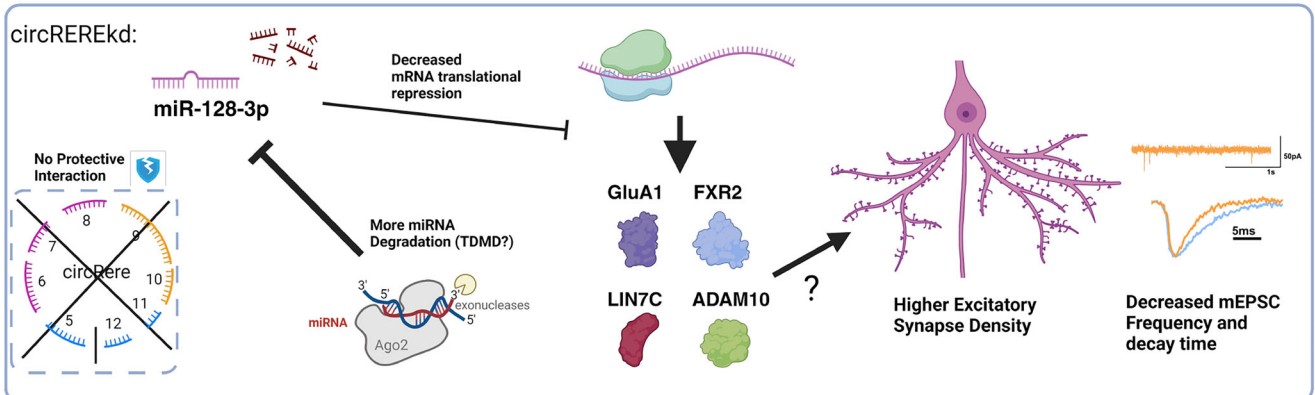

**Fig. 8** | **Proposed model of circRERE function.** Under basal conditions, *circRERE* maintains a dynamic protective interaction with miR-128-3p, preventing a reduction in the active miR-128-3p pool, e.g., potentially mediated by the target-dependent microRNA degradation (TDMD) pathway. High miR-128-3p activity, in turn, is necessary for efficient translational repression of synaptic genes involved in synapse formation and/or maturation. Upon *circRERE* kd, this protective interaction is lost, leading to enhanced miR-128-3p degradation, reduced translational repression of synaptic target genes, increased synapse density, and decreased mEPSC frequency. The miR-128-3p target genes mediating the effects on synapse density and mEPSC frequency are currently unknown. Created in BioRender. https://BioRender.com/v29u735.

## circRERE function in (patho-)physiology

Intriguing links between *circRERE* and neurodegenerative disorders have already been provided, in particular regarding Huntington's Disease (HD). For example, circRNA microarray profiling performed in a rat PC12-Q74 cell line identified 23 circRNAs as differentially expressed. Of particular note, 16 of the 19 downregulated circRNAs represent *circRERE* isoforms[23]. HD is also characterized by alterations in dopaminergic neurotransmission, which is perturbed in miR-128-3p deficient mice[51], thereby providing a link between *circRERE* and miR-128-3p. miR-128-3p further targets HD-associated genes, such as SP1, Huntington Interacting Protein 1 (Hip1) and Htt itself[58]. HD patients develop seizures, pertinent as reduced miR-128-3p is associated with increased neuronal activity[59]. Moreover, miR-128-3p levels have also

been shown as significantly downregulated in HD patients[60], transgenic HD monkeys[58], and transgenic HD mouse models[61]. Together, these results warrant further investigations into the functional relevance of the *circRERE*-miR-128-3p interaction for HD development.

Mutations within the *RERE* gene itself are associated with neurodevelopmental disorders. The "*RERE*-related neurodevelopmental syndrome" is characterized by global developmental delay, intellectual disability, hypotonia, seizures, and autism spectrum disorder[62]. Accordingly, *RERE* has been listed as an autism risk gene in the SFARI database[63] (https://gene.sfari.org/). *circRERE* levels have also been seen to increase in the hippocampus of a BTBR model of Autism[64]. To what extent *circRERE* dysregulation, e.g., caused by mutations in non-coding regions of the *RERE* gene, contributes to autism etiology is an

interesting topic for the future. More generally, given the over-representation of synaptic circRNA host genes in the SFARI database, circRNA dysregulation should be considered as a potential patho-physiological mechanism in autism and other synaptopathies.

# Methods

## Primary cell culture
Primary cortical and hippocampal neuronal cultures were prepared from embryonic day 18 (E18) male and female Sprague-Dawley rats (Janvier Laboratories). Euthanasia of pregnant rats for removal of embryonic brains was approved by the Veterinary Office of the Canton Zurich, Switzerland, under license ZH027/2021 and ZH112/2024. Pregnant Sprague-Dawley rats were housed under standard conditions with food and water ad libitum upon receipt. On the day of dissection, rats were euthanized by decapitation after anesthesia with isofluorane. All procedures were performed at 4 °C unless stated otherwise. The hippocampus from each embryo was dissected in precooled DM (HBSS,10 mM HEPES) and collected separately in 5 mL of DM. Once the hippocampi and cortex from all embryos were collected, tissue was dissociated by incubation with 500 µl of TrypLE Express (Thermo Fisher, 12604013) for 7 min in a 37 °C water bath, to aid dissociation hippocampi were mixed by inversion every 2 min during incubation time Afterward, hippocampi were washed twice with 5 mL of DM medium prewarmed at 37 °C. Mechanical cell dissociation was achieved by carefully triturating the hippocampi with a 1000 µl pipette tip in 500 µL of NBP+ (NBP/B27/Glutamax/Pen_strep medium). After dissociation, Hippocampal cells were counted diluted to a concentration of 120,000 cells/ml in NBP+ and plated. The same procedure was used for Cortical Neuron isolation. Dissociated cortical neurons were seeded on poly-L-ornithine-plated six-well plates (used for nucleofections), whereas hippocampal neurons were seeded on poly-L-lysine/laminin-coated coverslips in 24-well plates. Except for electrophysiology, all neuron cultures were maintained in Neurobasal plus (Thermo Fisher Scientific, A3582901) media supplemented with, 2 mM GlutaMAX, 2% B27 100 µg/ml streptomycin, and 100 U/ml penicillin (Invitrogen, Gibco) in an incubator with 5% $CO_2$ at 37 °C. Primary hippocampal neurons used for electrophysiology were maintained in Neurobasal-A (Thermo Fisher Scientific, 10888022) media supplemented with 2% B27, 2 mM GlutaMAX, 100 µg/ml streptomycin, and 100 U/ml penicillin (Invitrogen, Gibco).

## Transfection/nucleofection
All transfections of hippocampal cells were performed using Lipofectamine 2000 (Invitrogen), in duplicate/triplicate wells on DIV8/9 in Neurobasal plus medium (Thermo Fisher Scientific, A3582901), with the exception of electrophysiology experiments. 1 µg of total DNA was transfected per well in a 24-well plate, where an empty pcDNA3.1 vector was used to make up the total amount of DNA. Neurons were transfected in Neurobasal plus media in the absence of streptomycin and penicillin for 2 h, replaced with neuron culture media containing ApV (1:1000) for 45 min, which was washed away and replaced with conditioned media.

Hippocampal neuron transfections for electrophysiology were performed using Lipofectamine 2000 at DIV9 in Neurobasal-A medium (Thermo Fisher, 10888022). 1 µg of total DNA was transfected per well in a 24-well plate, where an empty pcDNA3.1 vector was used to make up the total amount of DNA. Before the addition of the lipofectamine/DNA mix, cells were equilibrated in warm Neurobasal-A containing ApV (1:1000) without penicillin and streptomycin for 30 min at 37 °C. Transfection incubation time was reduced to 1.5 h. Cells were then incubated with Neurobasal-A supplemented with ApV for 45 min, replaced with conditioned media, and maintained until the day of recording (DIV15/16). Transfections of primary hippocampal neurons utilized 7.5 ng of pSUPER constructs, 7.5 pmol/107.5 ng of siRNA, 150 ng GFP, and 300 ng of circRNA/miRNA overexpression

constructs. In the case of circRERESh, as described in the text, 2.5 ng of circRERE 1Sh, circRERE2Sh2, and circRERE3Sh pSUPER constructs was utilized instead of 7.5 ng of a single shRNA-expressing pSUPER plasmid.

Nucleofections were done on cortical neurons using the P3 Primary Cell 4D-Nucleofector X Kit (Lonza, LZ-V4XP-3024), on the day of preparation and dissociation (DIV0). 4 million dissociated cortical cells were electroporated with 3 µg total DNA per condition with the DC-104 program, seeded in six-well plates in DMEM/ GlutaMAX supplemented with 5% FBS and incubated for 4 h and then replaced with neuron culture media and incubated at 37 °C until harvesting. The following amounts of DNA were used for the relevant nucleofections: 2 µg pSUPER plasmid or circRNA OE plasmid, 1 µg GFP plasmid.

## circRNA reconstruction
Unique circRNA BSJs were reconstructed from RNA sequencing data from ref. 26 using the Deep computational Circular (DCC) RNA Analytics approach/circTest. Available at: https://github.com/dieterich-lab/DCC and https://github.com/dieterich-lab/CircTest.

## RNA interference
Silencer Select siRNA Custom RNAi Screen (Thermo Fisher) pools of 3× siRNAs against each investigated circRNA BSJ were generated, each staggered by 2–3 nt. Sense and antisense sequences for each siRNA are provided in the Supplementary Primer Table https://www.thermofisher.com/order/custom-genomic-products/tools/sirna/.

pSUPER shRNA-expressing plasmids (pSUPER Basic, Oligoengine, VEC-PBS0001/0002) producing19nt mature shRNA sequences were generated by oligo cloning between BglII and HindIII restriction enzyme sites and verified by sanger sequencing. Sense sequences are provided in the Supplementary Primer Table.

## circRNA Overexpression
Circular RNA overexpression utilized the Wilusz lab's pcDNA3.1(+) ZKSCAN1 MCS Exon Vector (Plasmid #69901, Addgene), with the sense sequence of the RERE exons 5–12 (ENSRNOT00000024443.4) cloned between EcoRV and SacII restriction enzyme sites by PCR extension addition of RE sites from randomly primed hippocampal rat cDNA, with the mutant synthesized by Geneart synthesis (Thermo Fisher). Cortical primary neurons at DIV0 were nucleofected with 2 µg of the OE or Mut plasmid and 1 µg GFP to confirm transfection efficiency. In primary hippocampal neurons, for synapse density experiments (Fig. 7), 300 ng OE or Mut plasmid was utilized for lipofection alongside 7.5 ng pSUPER shRNA plasmid, 150 ng GFP and made to 1 µg with empty pcDNA3.1 vector.

## miR-128-2 Overexpression
dsRed-miR-128-2 overexpression construct producing the chimeric intron of miR-128-2 and the control overexpression construct overexpression and control intron (dsRed-β-Actin) were kindly provided by Gregory Wulczyn (Charite Berlin), details provided in ref. 42.

## Immunocytochemistry, spine density, and image analysis
Stainings were performed on neurons fixed with 4% paraformaldehyde/4% sucrose/PBS. Fixation time was limited to 10–15 min to ensure intact postsynaptic compartments. For all imaging experiments, hippocampal cells were transfected on DIV8/9 in duplicate or triplicate wells in 24-well plates. For GluA1 surface staining, live cells were treated with primary antibody at 37 °C for 1 h. After washing the cells 4× times with fresh cell media, cells were fixed for 13 min with 4% paraformaldehyde/4% sucrose/PBS and washed with PBS. Coverslips were then transferred to a humidified chamber at room temperature, incubated in a secondary antibody in GDB for 1 h, washed with PBS, rinsed briefly with MilliQ water, and mounted onto glass slides for imaging. For PSD-95/Synapsin-1 co-staining or GluA1 total staining,

following 13-min 4% paraformaldehyde/4% sucrose/PBS fixation, the coverslips were incubated in GDB for 20 min. They were then immediately incubated at room temperature with primary antibodies diluted in GDB in the dark for 2 h and washed 4× times for 5–10 min. Secondary antibodies in GDB were applied for 1.5 h alongside Hoechst (1:2000). The following primary antibodies were used: Rabbit polyclonal anti-GluA1 (PC246 Calbiochem EMD Biosciences, at final concentrations 2 µg/ml for surface staining, 1:1000 for total staining), rabbit anti-Synapsin-1 (AB1543, 1:1,000; Merck Millipore), and mouse anti-PSD-95 (810401, 1:200; BioLegend). Alexa 488– 546– and 647–conjugated secondary antibodies (1:2000 dilution) were used for detection. Hoechst 33342 (Thermo Fisher) 1:2000 (DNA/Nuclear marker). Images are acquired with a confocal laser scanning microscope (Zeiss, CLSM 880) at 63× unless otherwise stated. Images were processed by Airyscan processing at 6.0 strength 3D, and maximum intensity projections of the Z-stacks were used for signal quantification. PSD-95/Synapsin co-cluster number and surface GluA1 particle number within cell area were analyzed with a custom-made Python script developed by D Colameo and can be added as a Plugin on Fiji (https://github.com/dcolam/Cluster-Analysis-Plugin) Zenodo. doi: 10.5281/zenodo.8167635. Fiji image analysis software is available at https://imagej.net/software/fiji/downloads. Transfected GFP plasmid produced a neuronal mask used to determine the inclusion of PSD-95/Syn1 Co-Clusters or Glua1 surface puncta within the neuron of interest. The number of puncta/Co-Clusters was then normalized to the area of the GFP mask referred to as "Synapse density" or "GluA1 Surface Puncta per square micron"[32]. Dendritic spines were also detected with our custom Fiji plugin based upon the structural characteristics of spines based on the GFP signal and intensity and the protrusion of the spine relative to its dendrite. Full details are available as CFG files for each experiment. In the case of the primary siRNA screen (Fig. 2), each biological replicate (N) was spread over 4 different 24-well plates due to the size of the screen. To account for this, each condition was normalized to a control condition on each plate (GFP) and then compared to siControl conditions (also GFP normalized). Immunocytochemistry analysis was performed blinded, primary screen blinding scheme and color coding noted in extended R markdown, available upon request.

### circRNA/mRNA FISH and miRNA FISH

Single-molecule (sm) FISH was performed using the ViewRNA Cell Plus kit according to the manufacturer's protocol (Thermo Fisher - 88-19000-99). circRNA FISH for *circStau2*, *circRMST*, and *circHomer1* instead utilized the ViewRNA Cell kit, with signal development with Fast Red Substrate, while *circRERE2* FISH utilized the ViewRNA Cell Plus kit with probe sequences provided in Supplementary Primer Table targeted against the respective BSJ's. For miR-128-3p miRNA FISH, an EDC cross-labeling step was added before the protocol (Thermo Fisher, 22980), using solutions of the Affymetrix QuantiGene ViewRNA miRNA ISH Cell Assay kit and following the manufacturer's described steps. HU-scramble-miR probe set (VM1-10338, Affymetrix) acted as a negative control condition. An additional Protease treatment was utilized for miRNA FISH; to preserve dendrite morphology, Protease QS was used at a dilution of 1:10,000 for 45 s. Rat Camk2a (VC4-15081) acted as a marker for excitatory neurons and allowed for the determination of cellular inclusion of other probe signals as CamkIIa fills the soma compartment and the proximal dendrites. hsa-miR-128-3p viewRNA Cell Plus probe (Assay ID:VM1-10249-VCP) allowed for detection of miR-128-3p signal. Note: miR-128-3p mature sequence is conserved between humans, mice, and rats.

### Sholl analysis

Images were taken on an Axio Observer Inverted Brightfield Microscope (Zeiss) using a 20× objective. The coverslip was subjected to tiling to reconstruct a high-resolution image of the full coverslip; this allowed for a more rigorous selection of representative cells for the overall coverslip. 200–250 tiles covered the majority of the coverslip, with an image resolution of ~35,000 × ~23,300 pixels, corresponding to an image size of ~7600 × ~5000 µm before cell selection. The level of dendritic arborization of pyramidal neurons was determined by Sholl analysis. This involved ten concentric circles being superimposed around the midpoint of the soma at 20um increments. The number of intersections across each circle was counted, allowing the Sholl profile of the dendritic arbor to be obtained. This process was automated with the FIJI "Deprecated Sholl" program. Nine/ten neurons for each condition were analyzed per independent biological replicate ($n = 3$). Note: ten neurons per condition with the exception of GFP N1/N2 with 9 neurons.

### Adult rat brain tissue collection

Adult Rat (female) was sacrificed by cervical dislocation, and various brain tissues were collected on an ice-cold glass plate and subsequently snap-frozen for RNA extraction (Trizol protocol) and gene expression analysis (qPCR) after Superscript III reverse transcription by random priming.

### RNA extraction and quantitative real-time PCR

RNA was isolated using RNA-Solv reagent (Omega Bio-tek) (tissue) or mirVana miRNA extraction kit (Cell culture). Genomic DNA was removed with TURBO DNAse enzyme (Thermo Fisher Scientific). Reverse transcription was performed using either the TaqMan MicroRNA Reverse Transcription Kit (Thermo Fisher Scientific) for miRNA detection or the Superscript III reverse transcriptase for mRNA/circRNA detection (Thermo Fisher) with random hexamer priming. qPCR was performed using either TaqMan Universal PCR Master Mix (Thermo Fisher Scientific) for microRNA detection or the iTaq SYBR Green Supermix with ROX (Bio-Rad) for mRNA/circRNA, and plates were read on the CFX384 Real-Time System (Bio-Rad). Data were analyzed via the ΔΔCt method and normalized to either U6 (for miRNAs) or GAPDH/Ywhaz/U6 (for mRNAs/circRNAs). mRNA/circRNA convergent (linear mRNA) and divergent (circRNA) primer information is indicated in the Supplemental Primer Table. TaqMan primers used were (Thermo Fisher Scientific): U6 snRNA (Assay ID: 001973), hsa-miR-128a-3p (Assay ID:002216).

### Luciferase assay miR-128-3p 2× perfect binding site reporters

Luciferase assays were performed using a dual-luciferase reporter assay (pmirGLO vector, Promega). 2× perfect binding sites for miR-128-3p were inserted into the firefly 3′UTR between the NheI/SalI restriction enzyme sites. Full sequence in Supplementary Primer Table. Triplicate conditions were transfected in primary HC neurons at DIV9, 50 ng of the pMirGlo plasmid of interest alongside 7.5 ng pSUPER construct or 300 ng miR-128 OE construct, 150 ng GFP and made up to 1 µg with pcDNA3.1. At DIV12/13, cells were lysed in Passive Lysis Buffer (diluted to 1×; Promega) for 15 min, and dual-luciferase assay was performed using homemade reagents (as described in ref. 32) on the GloMax Discover GM3000 (Promega).

### RNA sequencing culturing conditions/RNA Extraction

Cortical neuronal cultures after electroporation at DIV0 were maintained in Neurobasal plus medium supplemented with 2% B27, 3 mM GlutaMAX, 100 µg/ml streptomycin, and 100 U/ml penicillin (Thermo) in a 37 °C incubator with 5% $CO_2$ until lysis at DIV5. mirVana miRNA isolation kit with Total RNA Isolation Procedure was utilized for the extraction of total RNA. DNase treatment (Turbo DNase, Thermo Fisher) was performed to remove DNA contamination. The same RNA material was then used for PolyA and Small RNA sequencing (Novogene) as described.

## mRNA non-directional (polyA) sequencing (Novogene)

RNA sample was used for library preparation using NEB Next® Ultra RNA Library Prep Kit for Illumina®. Indices were included to multiplex multiple samples. Briefly, mRNA was purified from total RNA using poly-T oligo-attached magnetic beads. After fragmentation, the first strand of cDNA was synthesized using random hexamer primers, followed by the second strand of cDNA synthesis. The library was ready after end repair, A-tailing, adapter ligation, and size selection. After amplification and purification, the insert size of the library was validated on an Agilent 2100 and quantified using quantitative PCR (QPCR). Libraries were then sequenced on Illumina NovaSeq 6000 with PE150 according to results from library quality control and expected data volume. Quantification was performed using salmon 1.8.0 with --validateMappings on the mRatBN7.2 transcriptome. Surrogate variable analysis was performed with the sva package and two SVs used in the differential expression model. Features were filtered with edgeR::filterByExpr with min.count = 20, and differential expression was performed with edgeR's likelihood ratio test, comparing the knockdown to the two kinds of control samples taken together (the decision to pool the two control groups together was taken after observing the general lack of difference between them)[65].

## Small RNA-seq (Novogene)

The quality control procedures for total RNA include agarose gel electrophoresis (1%, 180 V, 16 min) and Nanodrop Spectrophotometer measurement, which are applied for preliminary monitoring of RNA degradation, concentration, and purity. Agilent 2100 Bioanalyzer is used to test RNA integrity and accurate quantification of RNA. After quality control checks, a small RNA library was prepared with the NEB Next® Multiplex Small RNA Library Prep Set for Illumina® according to the manufacturer's instructions.

Briefly, 3′ and 5′ adapters were ligated to 3′ and 5′ end of small RNA, respectively. Then the first strand cDNA was synthesized after hybridization with reverse transcription primer. The double-stranded cDNA library was generated through PCR enrichment. After purification and size selection, libraries with insertions between 18 and 40 bp were ready for sequencing with SE50. The constructed libraries underwent quality controls, including assessment of size distribution (Agilent 2100 Bioanalyzer) and molarity (quantified by qPCR). Qualified libraries were sequenced on an Illumina NovaSeq 6000 platform using S4 flow cells (Illumina, USA) using a SE50 bp mode. GEO accession ID: GSE261610. Short RNA analysis was performed with sports 1.0 with -M 1, using the authors' Rnor6 annotation. Reads were aggregated between genome- and library-aligned, and only miRNAs were considered for downstream analysis. Surrogate variable analysis was performed with the sva package and two SVs used in the differential expression model. Features were filtered with edgeR::filterByExpr with min.count = 30, and differential expression was performed for the polyA-RNA[66]

## GO term analysis/gene set enrichment analysis

Gene Ontology enrichment analysis (GO term) was performed using the TopGo algorithm (v.2.52.0)[67]. Available at https://bioconductor.org/packages/release/bioc/html/topGO.html.

Gene set enrichment analysis (GSEA) was conducted with the adaptive multilevel approach from the fgsea package v1.160[68] against mouse Hallmark and GO terms from the msigdbr package v7.5.1[69] using the logFC-signed −log10(p-value) of genes passing DEA-filtering (see above) as a signal, using the logFC-signed −log10(p-value) of genes passing DEA-filtering (see above) as a signal.

## Rolling circle amplification

Adult rat cortical RNA was subjected to reverse transcription with random hexamers and PCR-amplified using *circRERE2* BSJ-flanking or BSJ-spanning divergent qPCR primers (noted in Supplementary Primer List). Running PCR samples on TAE 1.5% Agarose gel electrophoresis revealed the expected size for the respective circRNA species.

## RNase R treatment/qPCR

Total RNA extract from adult rat cortex (2 µg) was incubated with 3 U/µg of RNase R (or mock-treated) at 37 °C for 10 min. Subsequently, the RNA was transferred back to ice and a 10% (200 ng) spike-in of E. coli total RNA was added. The RNA was re-extracted with acidic phenol-chloroform and ethanol-precipitated. The RNA concentration of the mock-treated sample was determined, and 1 µg was used for reverse transcription with Superscript III (Invitrogen) as described above, the same volume of the RNase R-treated RNA was used for reverse transcription. The E. coli spike-in was used for normalization in the qPCR with CysG primers.

## Electrophysiology

Whole-cell patch-clamp recordings were performed on an upright microscope (Olympus BX51WI) at room temperature. Data were collected with an Axon MultiClamp 700B amplifier and a Digidata 1550B digitizer and analyzed with pClamp11 software (all from Molecular Devices). Recording pipettes were pulled from borosilicate capillary glass (GC150F-10; Harvard Apparatus) with a DMZ-Universal-Electrode-Puller (Zeitz) and had resistances between 3 and 4 MW.

Miniature EPSCs (mEPSCs) were recorded from primary cultured hippocampal neurons on DIV15-16 after transfection in Neurobasal-A medium (10888022; Thermo Fisher Scientific) on DIV9. The extracellular solution (ACSF) was composed of (in mM) 140 NaCl, 2.5 KCl, 10 Hepes, 2 CaCl$_2$, 2 MgCl$_2$, 10 glucose (adjusted to pH 7.3 with NaOH), the intracellular solution of (in mM) 125 K-gluconate, 20 KCL, 0.5 EGTA, 10 Hepes, 4 Mg-ATP, 0.3 GTP, and 10 Na2-phosphocreatine (adjusted to pH 7.3 with KOH). For mEPSCs, 1 µM TTX and 1 µM Gabazine were added to the extracellular solution to block action-potential driven glutamate release and GABAergic synaptic transmission, respectively. Cells were held at −60 mV. The sampling frequency was 5 kHz, and the filter frequency was 2 kHz. Series resistance was monitored, and recordings were discarded if the series resistance changed significantly (≥10%) or exceeded 22 MΩ.

## ScanmiR

miRNA binding site prediction on reconstructed circRNA isoforms utilized the ScanMiR webtool version 1.5.2 https://ethz-ins.org/scanMiR/ based upon the Rat refseq Rn7 miRNA collection using default parameters as detailed in the interface[43].

## EnrichmiR

enrichMiR analyses and cumulative distribution (CD) plots were generated using the enrichMiR package. A CD plot was generated on the scanMiR annotation using the option to split by site affinity. The enrichment plot utilizes an aggregated statistical model with a significance-weighted average of the scaled enrichment scores. Repression scores were inverted, and scores were scaled by unit variance (across miRNAs); for each miRNA, the tests were weighted by the significance quantile within the test and then averaged across tests. Further information is available at enrichMiR publication. EnrichMir version 0.99.28 https://ethz-ins.org/enrichMiR/[41].

## Statistics and reproducibility

In our experimental design, we have technical replicates or plasmid mosaicism that are nested within true biological replicates, hence justifying the need to utilize generalized linear mixed models to test for fixed (true independent variable) effects while correcting for the variance introduced by random effects, i.e., additional sources of variance that are not directly related to the independent variable.

For the primary circRNA siRNA screen (Fig. 2), outputs (synapse co-cluster density, synapsin and psd-95 puncta density, and spine

density) were first tested for normality using the Shapiro–Wilk test. As they were not normally distributed, outputs were then normalized to the GFP condition, which served as a plate-specific transfection control, and then $\log_2$-transformed to decrease variance and approximate a normal distribution. A large sample-size per condition further justified the use of linear mixed-effects models as the distribution of sample means is normal.

To model the relationship between siRNA-mediated circRNA-knockdown and the outputs, a linear mixed effect model was utilized with transfection as the fixed effect and the nested plate types within each true biological replicate as the random effect. Post hoc multiple comparisons against a single siControl1 were conducted using Dunnett's procedure for multiple comparisons to a control. The same model specifications were used for synapse density, spine density, and volume analyses (an example of the model specification is provided below).

$$Model : lmer(\log 2(Synapse Density) \sim Transfection \\ + (1|Biological Replicate/Technical Replicate)$$

A similar linear mixed effect model was used to analyze synapse density upon shRNA mediated knockdown of *circRERE* isoforms (Fig. 3) and GluA1 surface puncta density and total expression (Fig. 4). For post hoc comparisons, Tukey multiple-testing adjustments for all pairwise comparisons were used for circRNA FISH quantification, secondary validation of *circRERE* knockdown, and both GluA1 receptor surface and total expression.

For the analysis of synapse density with *circRERE*sh and miR-128-3p overexpression (Fig. 6), a linear mixed-effects model with pairwise post hoc comparisons with multiple-testing correction using the Tukey method was used to calculate *p*-values, where the biological replicates are defined as a random effect. A similar linear model was also used to test for the effect of Transfection ("condition") on Synapsin and psd-95 puncta density, and spine density.

$$Model : lmer(Synapse\_Density \sim condition + (1|Replicate))$$

To analyze synapse co-cluster density upon transfection of combinatorial shRNA and miRNA overexpression constructs (Fig. 7), additional predictors were added to the general linear model to correct for the variance generated by mosaic plasmid expression and overexpression in an interaction test. The model specification is -

$$Model : lm(SynapseDensity \sim Experiment + extraPlasmid \\ + OE + KD^*isRescue)$$

Plots were generated in R using ggplot2. Violin plots were utilized due to the number of samples per condition, with embedded boxplots to visualize the sample distribution, including the median and interquartile range. Statistical analyses were conducted using linear mixed-effects model package *lme4*, and post hoc comparisons were conducted using the *emmeans* package. ImageJ/FIJI was used to adjust contrasts in microscope images as well as to insert scale bars. GraphPad Prism 9 was utilized for the plotting and statistical analysis of electrophysiology recordings, supplemental luciferase assays, and qPCRs utilizing either 1-way and 2-way ANOVAs or unpaired *T*-Tests. For Supplementary Figures, a detailed statistical analysis, exact "N", and biological and technical replicates are provided in source data files.

### Reporting summary
Further information on research design is available in the Nature Portfolio Reporting Summary linked to this article.

### Data availability
RNA sequencing data has been deposited to Gene Expression Omnibus (GEO, PolyA sequencing Fig. 5A, Small RNA sequencing Fig. 6A, accession number GSE261610). The data that support the findings of this study are available from the corresponding author upon request. Source data are provided with this paper.

### Code availability
FIJI pipeline templates utilized in this study for the analysis of synapse co-cluster and dendritic spine density are available at https://github.com/dcolam/Cluster-Analysis-Plugin. Zenodo: https://doi.org/10.5281/zenodo.8167635.[70]

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

## Acknowledgements

We greatly acknowledge the excellent technical assistance provided by Tatjana Wüst, Cristina Furler, and Roberto Fiore. We further thank David Colameo for the establishment and maintenance of FIJI-based image analysis pipelines and Michael Soutschek for assistance in GO term analysis and miRNA binding site illustration. We would like to thank visiting students: Alexandra Huber and Martin Breu for assistance with extended morphological analyses, Yu Wang for assistance with miRNA FISH analysis, and Connor Bitter and Julianna Strother for testing of miR-128OE constructs. We thank Gregory Wulczyn (Charite Berlin) for the gift of Mir-128-2-OE constructs. D.K. was supported by a PhD fellowship from the SNSF, NCCR RNA & Disease. G.S. received funding from the Deutsche Forschungsgemeinschaft (DFG; grant SPP1738 (SCHR1136/4-2)) and the Swiss National Science Foundation (SNSF; grants NeuroCirc (IZSTZ0_216044) and ALTRUISM (32NE30_189486)). C.D. received funding from the Deutsche Forschungsgemeinschaft (DFG; grant SPP1738 (DI 1501/5-2)).

## Author contributions

D.K. designed the project, prepared RNA samples, cloning, microscopy experiments, cell culture, treatments, stainings, RT-qPCR experiments, prepared figures and wrote the original manuscript draft. S.B. performed optimization of circRNA RNAi, circFISH, and primary knockdown screen. J.W. performed a patch-clamp recording of primary HC cultures. P.N. performed statistical analysis and generated plots for all histology experiments. P.L.G. performed RNA-seq analysis, statistical analysis of the total circRNA population in compartmentalized neurons and overlap with published datasets. C.D. reconstructed circRNA BSJs from RNA-seq data and performed statistical assessment. G.S. supervised the project, coordinated the collaboration, and wrote the manuscript.

## Competing interests

The authors declare no competing interests.
