## [Peer Review file · Nature Communications]

A functional screen uncovers circular RNAs regulating excitatory synaptogenesis in hippocampal neurons

Corresponding Author: Professor Gerhard Schratt

Version 0:

Reviewer comments:

Reviewer #1

(Remarks to the Author)

In this study, The group of G. Schratt examines the impact of circRNAs on synapse formation. They identify processes-enriched circRNAs from rat hippocampal primary neurons. They then perform a functional screen of selected candidates revealing that several circRNAs significantly impact synapse density. They subsequently focus on circRERE isoforms (hereafter referred to as "circRERE"). Unexpectedly, circRERE LOF leads to the reduction of excitatory synaptic transmission (reduced mEPSC frequency) despite a significant increase in synapse density. The concomitant reduction of GluA1 surface expression suggests the excess synaptic co-clusters may be functionally inactive. Next, the authors delve into the mechanism of action mediated by circRERE and provide a series of evidence indicating that circRERE isoform's effect on synapse formation is mediated by the direct binding and stabilization of miR-128-3p. In addition, they show that several miR-128-3p targets associated with glutamatergic synapse GO-term are also upregulated, although whether any of these mediate circRERE effects on synapse regulation remains unknown. Overall, this manuscript provides compelling evidence robustly demonstrating the involvement of a circRNA and associated isoforms in the regulation of excitatory synaptic transmission through direct miRNA stabilization. circRNAs have been known to be primarily associated with synapses although their roles have remained elusive. This study thus provides important novel insight into the function of circRNAs. I have highlighted below key issues, which, if addressed, would significantly strengthen the manuscript.

MAJOR COMMENTS:

1. A critical mechanistic link is missing in the manuscript to support their proposed model (Fig.8). While the authors show that miR-128-3p targets are upregulated upon circRERE LOF, in line with circRERE-mediated miR-128-3p stabilization, they do not provide evidence that any of the miR-128-3p targets are indeed involved in the circRNA-miRNA axis-mediated synapse density regulation. Furthermore, they even show that the miR-128-3p target *gria1*, a gene that encodes for glutamate receptor 1 (GluA1), is upregulated, but unexpectedly, the surface expression of GluA1 itself is shown to be downregulated. The authors argue that this is likely due to the concurrent increase in ADAM10, another miR-128-3p target, known to control the localization of GluA1 to synapses. Have the authors measured the global (in whole cells) protein expression of GluA1 to understand whether its level is overall increased? The authors should at the very least attempt to rescue the circRERE LOF-induced phenotype through the concomitant LOF of selected miR-128-3p targets in support of their model and general conclusions.

2. Related to point 1, the authors can't conclude in their abstract that "Mechanistically, circRERE knockdown resulted in a preferential upregulation of synaptic mRNAs containing binding sites for miR-128-3p *because* of a reduced protective interaction between miR128-3p and circRERE. *Accordingly*, overexpression of circRERE rescued exaggerated synapse formation[...]". They do not demonstrate a relationship of cause and effect so the words "because" and "accordingly" should be removed and their model in Fig. 8 revised (unless point 1 is addressed).

3. The authors identify 3 circRERE isoforms yet they do not consistently knock down or overexpress the same one for functional and mechanistic analyses. 3.1. The authors KD all three circRERE isoforms for the most part, based on the observation that: "a more robust increase in dendritic synapse density compared to the circRERE2 shRNA (Figure 3G) [is obtained], indicating that circRERE isoforms 1 and 3 contribute to the repression of excitatory synapse formation, albeit to a

lesser extent compared to circRERE2.” Yet they do not provide statistical evidence to support their claim (the distribution of the data for the two groups shown in Fig. 3G does not appear to be different to me). The authors do attempt to KD the 3 individual isoforms individually early in the study, and obtain significant change in synaptic density for circRERE1 (but it depends on the control used) but not for circRERE3 (see supplementary table linked to Fig2). This casts doubt on the validity of employing a broader LOF approach when all isoforms are KD together, as opposed to solely focusing on circRERE2 (see subsequent point). 3.2. The authors sometimes exclusively focus on circRERE2 KD as opposed to the KD of the three isoforms together (referred to in the ms as “circRERE”). For instance, circRERE2 shRNA is used to assess the impact on miR-128-3p target expression (Fig. 5) but circRERE shRNA reveals an impact on excitatory synapse transmission (Fig.4). This leaves one to wonder whether the data are consistent from one dataset to the next and why this strategy was adopted. 3.3. Importantly, it is puzzling to me why circRERE3 is used for overexpression in Fig. 7. The authors argue that all isoforms share the same miR-128-3p binding sites with the implication that they could be interchangeable but this is without taking into account the possible impact of the circRNA secondary structure, since circRERE3 and 2 differ in sequence. Possibly, the authors should try to keep a consistent approach throughout the ms (either KD of all isoforms, or focus on circRERE2).

4. Another puzzling aspect of the manuscript is the fact that the overexpression of circRERE has no impact on synaptic density. If circRERE stabilizes miR-128-3p stabilization, one would expect circRERE OE to also modulate synaptogenesis. How do the authors explain their results? Are miR-128-3p levels altered upon circRERE OE (e.g. Fig7B does not show MCS OE controls)?

5. The FISH for circRNAs results are incomplete and unclear. 5.1. In general, no negative controls nor unmerged/single channel images are shown with no clearly delineated neurons and dendrites, and cells are not cultured as monolayers. So it is hard to conclude from the presented pictures that any of the circRNA-associated signal is really present in neurites (could this be simply background signal?) (e.g. Fig. 1E, 3E, S5D, 6D). Fig.3E is particularly confusing. Split images and proper controls should be shown for these panels. 5.2. In S5D, circRERE2 appears in the somatic compartment mostly. Is a signal also detected in the nucleus? How come no signal is detected in the neurites while it is enriched in this compartment (Fig.1). 5.3. In Fig. 3E, the authors use a probe directed exclusively against linear RNA (exons 2-4, linRERE, red) or against both circRERE + linear RERE mRNA (exons 5-10; circ+lin RERE, grey). It is unclear whether the authors separately target Exon 5 and 10 which are in common with circRERE1, 2, 3 (therefore one can assume that the detected signal can be attributed to any of these circRNAs + linear RERE) or whether, as written in the method, they use one single “custom probe against RERE exons 5-10” - implying that they target the BSJ of circRERE1 (and thus not the linear RERE nor any other isoforms). This needs to be clarified. Similarly, it is unclear which probes are employed to target the linearRERE: are separate probes used for exon 2 and 4? If this is the case, exon 2 and 4 are also expressed by other circRERE isoforms and therefore do not constitute a proper mean for exclusive circRNA detection. This also needs to be clarified. Critically, the authors do not provide proper control to ascertain that the signal is indeed associated with circRNAs. For instance, does the signal decrease with circRERE shRNA? 5.4. For ISH-based miR-128 quantification in neurites, it is unclear from the picture how quantification of such low numbers of miRNA in neurites (median of 3 in controls decreased to 2 with shRNA) can be accurate, especially if one can't distinguish neurites from background or underlying cells. At the very least, controls and better illustrative pictures should be provided (see also point 6.1.). Is miR-128-3p level also reduced in the nucleus? Is the reduction in the processes occurring specifically at synapses?

6. Related to point 5, I find the scope of the paper somewhat confusing. They stipulate that “based on our results from RNA-sequencing, we decided to systematically characterize the function of process-enriched circRNAs in excitatory synaptogenesis.”. They undertake certain steps to assess whether process-enriched circRERE acts locally yet there is very little data to support this possibility. While circRERE isoforms are shown to be enriched in neurites as measured by RNAseq and qPCR, the ISH analysis overall does not support such enrichment. In addition, miR-128-3p appears to be particularly abundant in the nucleus (and soma compared to neurites) and the circRERE shRNA does not lead to the convincing reduction of miR-128-3p in neurites (see point 6.4. above) and the authors concluded that “despite its strong dendritic enrichment, circRERE knockdown affected miR-128-3p levels similarly in the dendritic and somatic compartment (Fig. 6F)”. miR-128-3p target GluA1 shows a concomitant reduction in cell surface expression instead of the expected increase. This overall leaves the impression that the goal was not reached and I wonder whether this aspect would be more suited for a separate, future manuscript.

7. I am puzzled by some of the graphs and related statistical analysis that is employed in the MS. 7.1. It is unclear to me why the authors often used a generalized linear mixed model (GLMM). This choice (over ANOVA for instance) should be justified. 7.2. It is also strange that sometimes, some controls do not have error bars (especially in the supplementary material e.g. Fig. 7D. Fig. S5B, S6 etc), as this would impact the outcome of the statistical analysis. 7.3. Furthermore, the authors do not always present the plots on synapse density in the same way (Fig. 2D vs Fig. 3G vs 6H vs 7D): sometimes it is normalized (e.g. 2D and 7D), sometimes it isn't. Non-normalized data should be consistently presented to be able to compare data from one figure to the next. 7.4. Critically, it is unclear to me why siRNA controls significantly impact synapse density (siRNA-control pool vs siRNA control 1 in Fig2C; siControl2-GFP vs siControl2-GFP in supplementary table (spine density). Was siRNA control used in toxic amount? Does siRNA control(s) similarly affect synapse density (compared to GFP) in Fig. 2D and S4A? 7.5. The authors often present heatmaps with replicates but no statistical analysis. This is particularly relevant for Fig.5E where the max log₂FC is relatively low (1) and where important mechanistic insight is concluded. Statistical analysis should also be presented.

8. Important controls are missing. 8.1. The authors should demonstrate that the overexpression construct does not generate

linear RNAs, an important possible confound. 8.2. In Fig. 7D, It would be important to examine whether the phenotype is (not) rescued with linear RERE as an additional control to be able to firmly confirm that it is the actual circularized RNA that is at play.

MINOR COMMENTS:

1. The word “process(es)-enriched” and not “dendritically-enriched” should be used throughout the results section. “Dendritic compartment” should not be used throughout.
2. The authors refer to “three structurally similar isoforms (circRERE1, circRERE2 and circRERE3)”. How do the authors define structurally similar isoform? For instance, one could argue that other isoforms are also structurally similar e.g. 5-9 or 4-10/11/12 etc.
3. Fig. 1 - there is virtually no information about the methods used to profile circRNAs and this should be added in the results (sequencing depth and read length) even though the RNAseq is derived from Colameo et al., 2021. How can the authors confidently determine the full length of the circRNAs knowing they do not employ nanopore long-read sequencing? This information should be added in the text of the result section.
4. Fig. 2A-B - PSD95 and Syn1 are difficult to distinguish in the figure – perhaps use different colours? It’s not straightforward to understand how synapse density was quantified and this should be clarified.
5. Fig. 3D - the author writes: “Over the course of rat hippocampal neurons differentiation in vitro, both linear and circular RERE isoforms initially decline, but peak at times of synapse formation (15-25 DIV; Fig. 3D).” To be able to make this claim, the authors would need to present replicates and appropriate statistical analysis (or remove this dataset). Here, it seems that only linRERE peaks. What is the circRNA to the linear ratio of circRERE isoforms throughout the course of synaptogenesis here?
6. Fig. 4I - it is unclear why negative values are obtained for GluA1 quantification, since they represent the number of puncta inside a specific area.
7. Fig 5D - what does the scale represent?
8. Fig. 6D - the figure is not representative of the data shown in Fig 6E. Pictures illustrating the data should be presented. Furthermore, it seems that neurons are overlapping giving the impression that a signal is present in a cell while it may be derived from a subjacent one (see also comment 6.1 above). How did the authors deal with these case scenarios to ensure that the quantification per cell was accurate?
9. Fig. 6G (and Fig 2B, Fig 3F, 4H, 6D, 6G, 7C) - the images illustrating change in dendritic spine densities are not all at the same scale which prevents a visual comparison. For each panel, zoomed pictures should be cohesively shown at the same scale to be able to compare them.
10. Fig. 6H/S3A/B - Statements should be backed up by statistical analysis. For instance related to Fig 6H: “Interestingly, miR-128OE alone did not further decrease synapse density, possibly due to its high expression already at basal levels.” is not assessed by statistical analysis comparing adequate groups. Similarly related to suppl. Fig. 3 A/B, the authors write: “we found that chronic Ptx treatment changed the expression of a distinct subset of circRNAs (circular-to-linear ratio, FDR<0.1) (suppl. Fig. 3 A/B) in a compartment-specific manner, e.g., circHomer1 (You et al., 2015) and several isoforms of the previously characterized circSlc8a1 (Hanan et al., 2020)”. This is not clearly shown by the data (e.g. there is no change in circHomer1 and circSlc8a1 (isoform 3) expression in the dendrites upon Ptx application). This statement should be toned down or backed up by clear statistical analysis.
11. Suppl. Fig 1 - the circularity of circRERE should also be validated with RNase R treatment.
12. Suppl Fig 4 - the outcome of the statistical analysis is missing on all panels. p-values should be shown.
13. Suppl Fig 5B, vs Suppl Fig 6B - is the linear counterpart of circRERE isoforms linRERE or linRERE2? These two supplementary figures do not agree.
14. Fig. S5D - do circRERE1 and 3 have a similar distribution as circRERE2? Does ISH reveal an enrichment in processes? Similarly in suppl. Fig. 5B, does rolling circle amplification demonstrate the circularity of circRERE1 and 3?
15. Suppl. Fig. 5E - the scale is not visible and it is not clear which RERE isoforms does the heat-map correspond to.
16. Suppl Fig 6A - the strategy employed to KD circRERE (circRERE1,2,3 isoforms) with shRNA is unclear. They do mention that they “simultaneously [knock] down circRERE isoforms 1-3” but which shRNA pool was selected from the results shown in Suppl Fig. 6A is not described and should be added in the MS. This is particularly relevant because 3 shRNA for circRERE2 is tested in Suppl Fig. 6. A schematic illustrating the shRNA used to KD circRERE 1 and 3 should also be added in Fig S6A.

17. The authors should consider using the correct nomenclature for circRERE1, 2, 3, the first time they introduce them in the text of the results (See Chen et al., 2021).

18. Discussion: the authors write: "We note however that absolute quantification of miR-128-3p by qPCR in primary hippocampal neurons led to an estimate of about ~4000 copies of miR-128-3p per neuron (de la Mata et al., 2015), which suggests that miRNA FISH is capturing only a subset of the total population (Fig. 6D-F). Thus, miR-128-3p is presumably present at sufficient quantity in dendrites to control the local expression of synaptic target mRNAs." I find this statement really far-fetched. I don't think the authors can strictly conclude about the sensitivity of their assay without experimentally determining copy number. The authors should be really cautious with their statement here and possibly remove it.

TYPOS:

1. Fig. 1F – x-axis is missing in one of the two graphs.

Reviewer #2

(Remarks to the Author)

The manuscript by Kelly, Schrott and colleagues presents evidence for a functional role of circRNAs produced by back-splicing of the RERE gene transcripts in synaptogenesis. The study is comprehensive and the paper is well written. Below are suggestions for improvement.

Major

1. Page 6. "Together, our bioinformatics results support a role for process-enriched circRNAs in synaptogenesis and synaptopathies, such as Autism Spectrum Disorder". This is an overstatement, please rephrase. The gene ontology enrichment data shows that genes encoding synaptic proteins are over-represented among those generating process-enriched circRNAs. These data do not demonstrate a role of process-enriched circRNAs in synaptogenesis (the rest of the paper does so for circRERE), and definitely does not provide evidence for their role in ASD.
2. Please add information on the statistical test for each p-value and exact p-value. eg Page 6: "miRNA binding sites are slightly more abundant in dendritically enriched circRNAs compared to the total population ($P < 0.044$)" What test? What is the exact p-value?
3. The first section of the paper is quite long, with some of the results rather vague or weak. I'd suggest removing some of the data that does not lead to clear conclusions such as: "We found that chronic Ptx treatment changed the expression of a distinct subset of circRNAs (circular-to-linear ratio, $FDR < 0.1$) (suppl. Fig. 3 A/B) in a compartment-specific manner, e.g., circHomer1 (You et al., 2015) and several isoforms of the previously characterized circSlc8a1 (Hanan et al., 2020). However we observe distinctive expression of circRNA BSJs (suppl. Fig. 3C) and the total exons of the circRNA tied to linear+circular counts, requiring further investigation to characterise their dynamic expression profiles (suppl. Fig. 3D)."
4. SFig3 lacks pvalues
5. Why does the siControlPool show significant downregulation in synapse density vs siControl1 in Fig2C? I would expect it to show no significant difference if there were no off-target effects.
6. Please add quantification to Fig 3E as the image on its own is not clear enough.
7. Page 10. "However, the knockdown of all 3 circRERE isoforms resulted in a more robust increase in dendritic synapse density compared to the circRERE2 shRNA (Figure 3G)" - there is no statistical test supporting this statement, so it's not clear if the difference is significant. Same comment for Fig4I and the corresponding text.

Minor

1. The labelling of Fig2C y-axis is confusing. Perhaps explain in the legend that it is an inverse log-scaled axis. Or else, just use $-\log_{10}(p\text{-value})$ which is common for Volcano plots.
2. Fig 5A please add the pvalue and FC thresholds to the legend.

Reviewer #3

(Remarks to the Author)

Dear Editor and Authors,

I appreciate the opportunity to review the manuscript titled "A functional screen uncovers circular RNAs regulating excitatory synaptogenesis in hippocampal neurons" by Darren Kelly et al. This paper presents a systematic investigation into the role of circular RNAs (circRNAs) in synaptogenesis. Circular RNAs have emerged as crucial regulators of gene expression, and their roles in neural development and function are becoming increasingly recognized as highly relevant in the field of neurobiology. Utilizing RNA interference, the authors identified several circRNAs that act as negative regulators of synapse formation and focus on circRERE, thus contributing to our understanding of circRNA-dependent microRNA regulation in synapse development and function. The study is well-executed, with robust experimental design and insightful discussion, and it gives a clear model for the proposed mechanism, making a significant contribution.

While I am very positive about the manuscript and its contribution to the field, I request some clarifications and additional details to enhance the clarity and comprehensiveness of the study.

Clarification on the Use of Different Neuron Types: On page 10, the manuscript mentions using rat cortical neurons for some experiments despite primarily focusing on hippocampal neurons. No further explanation is provided for this switch. Could the authors clarify the rationale behind using cortical neurons in these experiments?

Synapse and Dendritic Spine Analysis: I appreciate that the authors have reported synapse formation using both Syn/PSD95 co-labeling and GFP-labeled dendritic spine analysis. However, the exact numbers should be reported individually for PSD alone, Syn alone, and double-label synapses. This level of detail is essential for a clear understanding of the synapse distribution. Additionally, please clarify the criteria for selecting the counted segments, specifically regarding their distance from the soma, as spine density and GFP area depend highly on this parameter.

Number of Dendritic Spines in GluA1 Puncta Analysis: In the figure reporting the number of GluA1 puncta, the neurons are also labeled with GFP. I suggest including the number of dendritic spines in this same set of data.

Synapse Number in Figure 6: Please include the synapse number based on GFP-labeled dendritic spine analysis to provide a consistent metric for comparing different figures.

Figure Legends: The figure legends currently need more detail. Please include more information about the experimental design to help readers understand the data and its conclusions.

Typo in Figure 7: The term "GLLM" appears to be a typo. If I am correct, it should likely be "GLMM" (Generalized Linear Mixed Models).

Overall, I am very enthusiastic about this manuscript. It addresses a highly relevant topic and is of excellent quality. With these minor revisions, I strongly recommend its publication.

Thank you for considering these suggestions. Addressing these points will further strengthen the manuscript and provide additional clarity for readers.

Version 1:

Reviewer comments:

Reviewer #1

(Remarks to the Author)

The authors have adequately replied to the vast majority of my comments / suggestions. On a few occasions, the authors mention new data that are actually not shown, so I can't assess the reply to these major comments (see below):

Q1: "We observed that ADAM10 inhibition led to a reduction in synapse co-clusters under control conditions and normalized synapse co-cluster density under circRERE KD conditions." Here the data is not included in the reply to reviewers.

Q3.1: "Using a miR-128-3p perfect binding site luciferase reporter assay (Fig. 6C), we found that knockdown of both circRERE-1 and 2, but not-3 led to a significant decrease in miR-128-3p activity." Such data is now shown in Fig. 6C.

Q3.1. "we consistently observe a trend towards bigger effect sizes with the circRERE shRNA pool targeting all 3 major circRERE isoforms compared to circRERE-2 alone (e.g., synapse density (Fig. 3G), GluA surface expression (Fig. 4H), miR-128-3p activity (Fig. 6C))," Fig. 4H and 6C does not compare circRERE shRNA with circRERE-2 shRNA alone.

Reviewer #2

(Remarks to the Author)

The authors have addressed all of my questions, thank you!

Reviewer #3

(Remarks to the Author)

Dear Editor and Authors,

As the reviewer for the manuscript titled "A functional screen uncovers circular RNAs regulating excitatory synaptogenesis in hippocampal neurons," I am pleased to confirm that the authors have adequately addressed all the concerns and suggestions raised in the first revision.

The authors have satisfactorily addressed all minor points, such as typographical errors and formatting issues. Additionally, they have effectively resolved the significant points raised during the review. Briefly, they clarified the rationale for using rat cortical neurons in a subset of experiments that required high cell numbers. They also provided detailed analyses of Synapsin-1, PSD-95 puncta, and dendritic spine density and explained their unbiased approach to selecting dendritic segments for analysis. Additionally, new analyses of dendritic spine density in GluA1 datasets have been included, and the

rationale for excluding results from live-stained samples due to compromised morphology was convincingly explained.

The revised manuscript is of high quality, and the authors have made an exceptional effort to address all my concerns. I am very enthusiastic about this manuscript and strongly recommend its publication.

Sincerely,

Dario Maschi, Ph.D.
Assistant Professor
Department of Psychiatry
Washington University in St. Louis
4444 Forest Park Ave | St. Louis, MO 63108
Box 8134 | Suite 5120
St. Louis, MO 63108
Email: dario.maschi@wustl.edu

Version 2:

Reviewer comments:

Reviewer #1

(Remarks to the Author)

The authors have answered all of my comments. I congratulate them on their insightful study.

We are extremely grateful to the reviewers for their positive feedback and constructive comments, which helped us to further improve the quality of our manuscript. Please find below a detailed point-by-point response to the individual comments (in red).

Reviewer #1 (Remarks to the Author):

In this study, The group of G. Schratt examines the impact of circRNAs on synapse formation. They identify processes-enriched circRNAs from rat hippocampal primary neurons. They then perform a functional screen of selected candidates revealing that several circRNAs significantly impact synapse density. They subsequently focus on circRERE isoforms (hereafter referred to as “circRERE”). Unexpectedly, circRERE LOF leads to the reduction of excitatory synaptic transmission (reduced mEPSC frequency) despite a significant increase in synapse density. The concomitant reduction of GluA1 surface expression suggests the excess synaptic co-clusters may be functionally inactive. Next, the authors delve into the mechanism of action mediated by circRERE and provide a series of evidence indicating that circRERE isoform's effect on synapse formation is mediated by the direct binding and stabilization of miR-128-3p. In addition, they show that several miR-128-3p targets associated with glutamatergic synapse GO-term are also upregulated, although whether any of these mediate circRERE effects on synapse regulation remains unknown. Overall, this manuscript provides compelling evidence robustly demonstrating the involvement of a circRNA and associated isoforms in the regulation of excitatory synaptic transmission through direct miRNA stabilization. circRNAs have been known to be primarily associated with synapses although their roles have remained elusive. This study thus provides important novel insight into the function of circRNAs. I have highlighted below key issues, which, if addressed, would significantly strengthen the manuscript.

MAJOR COMMENTS:

1. A critical mechanistic link is missing in the manuscript to support their proposed model (Fig.8). While the authors show that miR-128-3p targets are upregulated upon circRERE LOF, in line with circRERE-mediated miR-128-3p stabilization, they do not provide evidence that any of the miR-128-3p targets are indeed involved in the circRNA-miRNA axis-mediated synapse density regulation. Furthermore, they even show that the miR-128-3p target *gria1*, a gene that encodes for glutamate receptor 1 (GluA1), is upregulated, but unexpectedly, the surface expression of GluA1 itself is shown to be downregulated. The authors argue that this is likely due to the concurrent increase in ADAM10, another miR-128-3p target, known to control the localization of GluA1 to synapses.

Have the authors measured the global (in whole cells) protein expression of GluA1 to understand whether its level is overall increased?

As suggested by this reviewer, we have now measured the total levels of GluA1 protein in fixed primary hippocampal neurons that had been transfected with either control or circRERE shRNA constructs using immunocytochemistry (new Fig. 4X). This analysis revealed that total GluA1 levels were significantly increased in circRERE shRNA

transfected neurons, consistent with our results from RNA-seq (Fig. 5E) and in support of GluA1 being a direct target of miR-128-3p downstream of circRERE. Altogether, this result suggests that circRERE, in addition to GluA1, controls gene(s) involved in GluA1 surface trafficking such as ADAM10. Further experiments beyond the scope of this project are needed to elucidate the molecular mechanisms underlying the different effect on GluA1 expression/localization upon circRERE KD.

The authors should at the very least attempt to rescue the circRERE LOF-induced phenotype through the concomitant LOF of selected miR-128-3p targets in support of their model and general conclusions.

As suggested by this reviewer, we have now attempted to rescue the circRERE LOF-induced phenotype (increase synaptic co-cluster density) through concomitant LOF of one of the miR-128-3p targets, ADAM10. ADAM 10 was chosen since it has been shown previously to be involved in excitatory synapse formation, and a highly specific pharmacological inhibitor is available for LOF without the need of expressing additional constructs. We observed that ADAM10 inhibition led to a reduction in synapse co-clusters under control conditions and normalized synapse co-cluster density under circRERE KD conditions. This result indicates that increased ADAM 10 expression/activity is an important event downstream of circRERE knockdown in the regulation of excitatory synapse formation. While this result is encouraging, we feel that many additional experiments will have to be performed to define the function of ADAM10 in this pathway, especially with regard to the different roles of circRERE in synapse formation and functional maturation. Such follow-up studies will form the basis of an entirely new manuscript and are clearly beyond the scope of this paper, whose focus is on the circRERE/miR-128-3p interaction. We therefore decided not to include the ADAM10 data in our revised manuscript.

2.Related to point 1, the authors can't conclude in their abstract that "Mechanistically, circRERE knockdown resulted in a preferential upregulation of synaptic mRNAs containing binding sites for miR-128-3p *because* of a reduced protective interaction between miR128-3p and circRERE. *Accordingly*, overexpression of circRERE rescued exaggerated synapse formation[...]" . They do not demonstrate a relationship of cause and effect so the words "because" and "accordingly" should be removed and their model in Fig. 8 revised (unless point 1 is addressed).

According to the suggestion of the reviewer, we have now toned down this statement in the abstract of the revised paper.

3.The authors identify 3 circRERE isoforms yet they do not consistently knock down or overexpress the same one for functional and mechanistic analyses.

3.1. The authors KD all three circRERE isoforms for the most part, based on the observation that: "a more robust increase in dendritic synapse density compared to the circRERE2 shRNA (Figure 3G) [is obtained], indicating that circRERE isoforms 1 and 3 contribute to the repression of excitatory synapse formation, albeit to a lesser extent compared to circRERE2." Yet they do not provide statistical evidence to support their claim (the distribution of the data

for the two groups shown in Fig. 3G does not appear to be different to me). The authors do attempt to KD the 3 individual isoforms individually early in the study, and obtain significant change in synaptic density for circRERE1 (but it depends on the control used) but not for circRERE3 (see supplementary table linked to Fig2). *This casts doubt on the validity of employing a broader LOF approach when all isoforms are KD together, as opposed to solely focusing on circRERE2 (see subsequent point).*

To address this concern, we have now carefully re-assessed the contribution of the three circRERE isoforms. Using a miR-128-3p perfect binding site luciferase reporter assay (Fig. 6C), we found that knockdown of both circRERE-1 and 2, but not-3 led to a significant decrease in miR-128-3p activity. This result is consistent with our data from the functional screen, where we observed significant changes in synapse density upon knockdown of circRERE-2 and (to a lesser extent) circRERE-1. The different effectiveness of the knockdowns further correlates with the RNA-seq data, which shows highest expression for circRERE-2, followed by circRERE-1 and -3 (Fig. 3B). In further support of a functional contribution of circRERE-1, we consistently observe a trend towards bigger effect sizes with the circRERE shRNA pool targeting all 3 major circRERE isoforms compared to circRERE-2 alone (e.g., synapse density (Fig. 3G), GluA surface expression (Fig. 4H), miR-128-3p activity (Fig. 6C)), although, as correctly pointed out by this reviewer, this usually didn't reach statistical significance. Overall, we are convinced that these results provide a strong rationale for using a circRERE shRNA pool for most of the functional experiments.

3.2. The authors sometimes exclusively focus on circRERE2 KD as opposed to the KD of the three isoforms together (referred to in the ms as "circRERE"). For instance, circRERE2 shRNA is used to assess the impact on miR-128-3p target expression (Fig. 5) but circRERE shRNA reveals an impact on excitatory synapse transmission (Fig.4). This leaves one to wonder whether the data are consistent from one dataset to the next and why this strategy was adopted.

The strategy was adopted since circRERE-2 originally emerged as the top candidate from the screen, but later experiments (see above) revealed also a contribution of circRERE-1 to miR-128-3p regulation (See our response to point 3.1). Therefore, we gradually switched to the circRERE shRNA pool over the course of the project. We still want to stress that the biological effects of circRERE2 KD and circRERE KD are highly comparable for all readouts we tested, and therefore do not see the necessity to retrospectively repeat experiments where only circRERE-2 KD was used (e.g., RNA-seq).

3.3. Importantly, it is puzzling to me why circRERE3 is used for overexpression in Fig. 7. The authors argue that all isoforms share the same miR-128-3p binding sites with the implication that they could be interchangeable but this is without taking into account the possible impact of the circRNA secondary structure, since circRERE3 and 2 differ in sequence. Possibly, the authors should try to keep a consistent approach throughout the ms (either KD of all isoforms, or focus on circRERE2).

We chose circRERE-3 overexpression, since we wanted to make sure to include all exons (Also the ones downstream of the miR-128-3p binding sites) which could possibly affect the rescue capability. Although endogenous circRERE-3 appears to contribute only to a minor extent to miR-128-3p stabilization (Fig. 6C), this is likely

due to its low expression in hippocampal neurons. Otherwise, we have no indication that the extra exon present in circRERE-3 might have any inhibitory function. In fact, we observe an almost complete rescue of the circRERE knockdown synapse phenotype by overexpressing circRERE-3, which argues that it is in principle fully capable of miR-128-3p stabilization. This is further supported by the observation that the circRERE-3 rescue capability is entirely dependent on the presence of the miR-128-3p binding sites. While in retrospect, a more consistent strategy focusing on circRERE-2 might have been desirable, we do not feel that performing additional experiments with overexpression of different circRERE isoforms would provide any important extra information regarding circRERE function in miR-128-3p stabilization and synapse regulation.

4. Another puzzling aspect of the manuscript is the fact that the overexpression of circRERE has no impact on synaptic density. If circRERE stabilizes miR-128-3p, one would expect circRERE OE to also modulate synaptogenesis. *How do the authors explain their results? Are miR-128-3p levels altered upon circRERE OE (e.g. Fig 7B does not show MCS OE controls)?*

We have now added the requested control (**new Fig. 7B**), which shows that overexpression of circRERE-3 wild-type, but not a miR-128-3p binding site mutant, leads to increased miR-128-3p expression, consistent with a stabilizing function. Since miR-128-3p is one of the most highly abundant microRNAs in rat primary hippocampal neurons, we speculate that increasing miR-128-3p levels further above baseline has no discernable effect on synapse formation, presumably since high baseline miR-128-3p levels are sufficient to effectively bind and repress all important high-affinity targets of miR-128-3p. This interpretation is also supported by our observation that miR-128-3p overexpression alone, similar to circRERE-3 OE, has no significant effect on synapse density (Fig. 6H). A respective statement has been added to the revised manuscript on page 13 (“Interestingly, miR-128OE alone did not further decrease synapse density, possibly due to its high expression already at basal levels”).

5. The FISH for circRNAs results are incomplete and unclear.

5.1. In general, no negative controls nor unmerged/single channel images are shown with no clearly delineated neurons and dendrites, and cells are not cultured as monolayers. *So it is hard to conclude from the presented pictures that any of the circRNA-associated signal is really present in neurites (could this be simply background signal?)* (e.g. Fig. 1E, 3E, S5D, 6D). *Fig. 3E is particularly confusing. Split images and proper controls should be shown for these panels.*

We politely disagree with this reviewer that cells were not cultured as monolayer. We employ a protocol that is routinely used in the neuroscience field for decades whereby neurons are plated as single cells on PDL/laminin-coated glass coverslips (e.g., Pacifici&Peruzzi, J. Vis. Exp. 2012; doi: [10.3791/3965](https://doi.org/10.3791/3965)). During the development, they will form synaptic connections, but remain as a single layer. The use of confocal imaging results in stacks of individual optical sections, which in turns allows us to unambiguously assign fluorescent signals (e.g., derived from circRNA-specific

probes) to specific neuronal compartments (e.g., nucleus, soma, dendrites) within individual cells (**new suppl. Fig. 5**). Therefore, we can confidently quantify circRNA-associated signals in the somatic and process compartment of individual neurons. In addition, as requested by the reviewer, we now show unmerged/single channel images for the FISH experiments and delineate neuronal soma and dendrites (**new Fig. 3D**). Regarding specificity of the signal, we have now also performed circRERE FISH in the presence of circRERE knockdown and added a scrambled control probe (**new Fig. 3D, E**; See our comment below for more detail).

5.2. In S5D, circRERE2 appears in the somatic compartment mostly. Is a signal also detected in the nucleus? How come no signal is detected in the neurites while it is enriched in this compartment (Fig.1).

We thank the reviewer for raising the important point of RNA enrichment in specific cellular compartments. First, we would like to clarify the terminology “process enriched or process enrichment”. This term always refers to a relative enrichment, e.g., in the case of the qPCR results from Fig. 1D, relative to the Gapdh mRNA. Furthermore, since equal amounts of RNA from the different compartments were used for RT qPCR, but the process-compartment contains much less total RNA compared to the somatic/nuclear compartment at the individual cell level (10-15%; Perez et al., eLife 2021), this calculation systematically overestimates the absolute amount of a given RNA in the process compartment. Therefore, even for a classical “dendritic mRNA”, like Camk2alpha, FISH would show less puncta in dendrites compared to the soma/nucleus, although RT-qPCR analysis indicates a two-fold relative enrichment in the process-compartment (Fig. 1D). While we therefore do not expect to see more puncta of circRERE-2 (or any other “process-enriched” circRNA) in dendrites compared to the soma in absolute terms, additional factors might contribute to the scarcity of the circRERE-2 signal in general. Foremost, the circRERE-2 probe is likely suboptimal due to an unfavorable sequence surrounding the unique circRERE-2 BSJ. This view is supported by our observation that the same smFISH protocol worked quite efficiently with probes directed against the BSJ of other process-enriched circRNAs, e.g., circHomer1, circRMST and circStau2 (Fig. 1D). In addition, circRERE might be associated with protein complexes, which are resistant to the mild protease treatment which is included in our smFISH protocol, leading to reduced accessibility for FISH probes and reduced detection. This argument is further supported by our recent results, which show a highly improved recovery of circRERE signal using the BSJ-specific FISH probe by simply increasing the duration of protease treatment (**new Fig. 3D, E**).

Given all these considerations, we have added more discussion on the meaning of “process-enrichment” (p. 16f.) and decided to tone down our statements regarding a specific dendritic function of the circRERE/miR-128 pathway during synaptogenesis throughout the manuscript.

5.3. In Fig. 3E, the authors use a probe directed exclusively against linear RNA (exons 2-4, linRERE, red) or against both circRERE + linear RERE mRNA (exons 5-10; circ+lin RERE, grey). It is unclear whether the authors separately target Exon 5 and 10 which are in common with circRERE1, 2, 3 (therefore one can assume that the detected signal can be attributed to any of these circRNAs + linear RERE) or whether, as written in the method, they use one single “custom probe against RERE exons 5-10” - implying that they target the BSJ of

circRERE1 (and thus not the linear RERE nor any other isoforms). This needs to be clarified. Similarly, it is unclear which probes are employed to target the linearRERE: are separate probes used for exon 2 and 4? If this is the case, exon 2 and 4 are also expressed by other circRERE isoforms and therefore do not constitute a proper mean for exclusive circRNA detection. This also needs to be clarified. *Critically, the authors do not provide proper control to ascertain that the signal is indeed associated with circRNAs. For instance, does the signal decrease with circRERE shRNA?*

We largely agree with the concerns of this reviewer regarding the specificity of our original FISH approach for circRERE. Given these caveats, we have now removed the corresponding panels entirely and instead further improved FISH with the highly specific BSJ-targeting probe for the dominant isoform circRERE-2, which in particular led to increased signal recovery in dendrites (**new Fig. 3D, E**). Nevertheless, even with this optimized protocol, we are on average able to detect only about 10 puncta/cell, only a few of which are localized to processes (see also our discussion on point 5.2). Importantly, we now also provide the requested control conditions, showing a significant reduction of circRERE signal upon co-transfection of the circRERE shRNA and greatly reduced signal intensity when using a scrambled FISH probe (**new Fig. 3D, E**). Altogether, this provides compelling evidence that circRERE-2 is present in both the somatic and dendritic compartment of primary rat hippocampal neurons.

5.4. For ISH-based miR-128 quantification in neurites, it is unclear from the picture how quantification of such low numbers of miRNA in neurites (median of 3 in controls decreased to 2 with shRNA) can be accurate, especially if one can't distinguish neurites from background or underlying cells. *At the very least, controls and better illustrative pictures should be provided (see also point 6.1.). Is miR-128-3p level also reduced in the nucleus? Is the reduction in the processes occurring specifically at synapses?*

As requested by this reviewer, we have now included an additional “no probe” control condition and better illustrative pictures which demonstrate the specificity of our FISH signals (**new Fig. 6D-F**). We are highly confident that we can faithfully distinguish signal present within neuronal dendrites from signal outside the neurons and noise by our confocal microscopy approach (**new suppl. Fig. 5**; please also refer to our answers regarding point 5.1 above). The same applies for distinguishing puncta present in the soma from those in the nucleus (most puncta which appear nuclear in fact are located above the nucleus in the somatic compartment). Our results are further consistent with those of a previous study which described miR-128-3p as a synaptic miRNA based on RNA-seq from microfluidics chambers and synaptosomes (Epple et al., 2021; cited in the paper). However, making further statements about synaptic localization would require co-staining for synaptic marker proteins, which is currently not compatible with our smFISH protocol due to a necessary protease treatment step which will destroy the corresponding antibody epitopes.

6. Related to point 5, I find the scope of the paper somewhat confusing. They stipulate that “based on our results from RNA-sequencing, we decided to systematically characterize the function of process-enriched circRNAs in excitatory synaptogenesis.”. They undertake certain

steps to assess whether process-enriched circRERE acts locally yet there is very little data to support this possibility. While circRERE isoforms are shown to be enriched in neurites as measured by RNAseq and qPCR, the ISH analysis overall does not support such enrichment. In addition, miR-128-3p appears to be particularly abundant in the nucleus (and soma compared to neurites) and the circRERE shRNA does not lead to the convincing reduction of miR-128-3p in neurites (see point 6.4. above) and the authors concluded that “despite its strong dendritic enrichment, circRERE knockdown affected miR-128-3p levels similarly in the dendritic and somatic compartment (Fig. 6F)”. miR-128-3p target GluA1 shows a concomitant reduction in cell surface expression instead of the expected increase. This overall leaves the impression that the goal was not reached and I wonder whether this aspect would be more suited for a separate, future manuscript.

We agree with this reviewer that our current data does not support an explicit function of the circRERE/miR-128 regulation in the synapto-dendritic compartment (See also our comments above). However, the documented presence of both molecules in dendrites (see our comments to point 5.4.) leaves open the possibility that they are also engaged in the local control of synaptic protein synthesis, in addition to a somatic contribution. To tease apart the individual contributions of the somatic and dendritic pools of circRERE/miR-128 to excitatory synaptogenesis will require sophisticated additional experimentation which in our view is beyond the scope of the present manuscript. This could for example involve a recently designed tool which purportedly is able to knockdown circRNAs and lncRNAs specifically in the synapto-dendritic compartment (Liau et al., Nat. Comm. 2023). In the meantime, we have toned down our statements regarding an exclusive synapto-dendritic function of circRERE/miR-128 throughout the manuscript (e.g., p. 13) and provided further discussion (p. 19f.).

The overarching goal of the study, namely the elucidation of circRNAs which regulate excitatory synaptogenesis, however has been clearly reached in our view. We further provide a specific mechanism, which includes the stabilization of a highly expressed microRNA and the concomitant reduction of microRNA targets involved in the synapse regulation.

7.I am puzzled by some of the graphs and related statistical analysis that is employed in the MS.

7.1. It is unclear to me why the authors often used a generalized linear mixed model (GLMM). This choice (over ANOVA for instance) should be justified.

We used GLMMs whenever the data had a nested (i.e. multilevel) structure, as is recommended (see e.g. Aarts et al., Neuron 2014, <https://www.nature.com/articles/nn.3648>). For example, different neurons or fields of view of the same culture (e.g. as in Figure 2) cannot be treated as independent samples (which a simple ANOVA would do), otherwise this would lead to increased type I error. GLMMs function essentially like an ANOVA but accounting for sample-level variations as random effects.

7.2. It is also strange that sometimes, some controls do not have error bars (especially in the supplementary material e.g. Fig. 7D. Fig. S5B, S6 etc), as this would impact the outcome of the statistical analysis.

We have now consistently added error bars and adjusted the statistical analysis throughout the manuscript.

7.3. Furthermore, the authors do not always present the plots on synapse density in the same way (Fig. 2D vs Fig. 3G vs 6H vs 7D): sometimes it is normalized (e.g. 2D and 7D), sometimes it isn't. Non-normalized data should be consistently presented to be able to compare data from one figure to the next

We have now consistently presented non-normalized data (Fig. 3G, 6H, 7D), except for Fig. 2D. In this large-scale screening experiment, we had to use multiple plates to cover all siRNA conditions. Each of these plates contained its own GFP control conditions, which was used for normalization to account for a potential variability in the cell status between the different plates. We can show that normalizing to the GFP control in fact reduces the variability of the data as compared to non-normalized data in this specific case (**new suppl. Fig. 3F**).

7.4. Critically, it is unclear to me why siRNA controls significantly impact synapse density (siRNA-control pool vs siRNA control 1 in Fig2C; siControl2-GFP vs siControl2-GFP in supplementary table (spine density)). Was siRNA control used in toxic amount? Does siRNA control(s) similarly affect synapse density (compared to GFP) in Fig. 2D and S4A?

The reason for the negative effect of the siRNA control pool on synapse density is currently unclear. The total amount used for this concentration was the same as for individual siRNAs (siControl1, siControl2), ruling out toxicity due to excess siRNA delivery. A plausible explanation could be that only a specific combination of siRNA sequences exerts a toxic effect, but this is difficult to test experimentally.

We want to stress however that siControls had no significant effect compared to the GFP only condition for the majority of readouts (suppl. Fig. 3A-C). Moreover, the vast majority of the siRNA pools directed against specific circRNAs also did not show any adverse effects, thereby serving as additional “negative controls”. Altogether, we are highly confident that the identified circRNAs represent true “positive hits”, especially since they all showed a relative increase in synapse density compared to control conditions which is not consistent with any non-specific toxicity. Finally, at least two of the positive hits could be further validated with independent shRNA-expressing plasmids.

7.5. The authors often present heatmaps with replicates but no statistical analysis. This is particularly relevant for Fig.5E where the max log₂FC is relatively low (1) and where important mechanistic insight is concluded. Statistical analysis should also be presented.

We think that there is a misunderstanding regarding the data presented in Fig. 5E. All genes presented in this heatmap are in fact differentially expressed between circRERE shRNA-2 and control conditions with an FDR<0.05 (see also Fig. 5A). We

therefore don't see the need for the presentation of additional statistical analysis in this case.

8. Important controls are missing. 8.1. *The authors should demonstrate that the overexpression construct does not generate linear RNAs, an important possible confound.*

To address this point, we have now quantified circular and linear transcripts originating from overexpression of circRERE-3 by qPCR using primer sets which either detect exclusively the overexpressed circular RNA (circRERE-BSJ) or a combination of circular and linear RERE transcripts (both overexpressed and endogenous; circRERE-internal). Thereby, we observed that the ratio of circular RERE/total RERE was about 0.8 – 0.9 in neurons overexpressing either circRERE WT or circRERE MUT, indicating that the vast majority of overexpressed RERE transcripts are circular (**new suppl. Fig. 11 D-F**). More importantly, this ratio did not significantly change upon RNaseR treatment, suggesting that RERE transcripts which are exclusively detected by the circRERE-internal primer set might in fact not represent linear isoforms, but rather endogenous circRERE species. Together, this data demonstrates that the contribution of linear RERE to the effects of circRERE overexpression is very small and probably negligible.

8.2. *In Fig. 7D, It would be important to examine whether the phenotype is (not) rescued with linearRERE as an additional control to be able to firmly confirm that it is the actual circularized RNA that is at play.*

We agree with this reviewer that knowing whether circularity of the RERE sequence is required for mir-128 stabilization and the subsequent effects on synaptogenesis is an interesting aspect of the study. However, as outlined in more detail below, several caveats would make the interpretation of a rescue experiment with a linear RERE transcript as suggested by this reviewer extremely difficult. In conjunction with the expected complexity of the experiments, which would require at least 3 months for completion, we decided not to perform it. First, the absence of a rescue by overexpression of a truncated linear RERE transcript encompassing exons 5-12 would not unequivocally demonstrate a requirement for circularity, but could also originate from ineffective expression, instability, aberrant cellular localization, or non-specific interactions by such an isoform which are together hard to control for. Second, even if we would observe a rescue by the truncated linear RERE isoform, this also doesn't necessarily prove the non-functionality of circular RERE, especially since such a truncated linear RERE isoform is normally not expressed in neurons and therefore also cannot account for the miR-128-3p stabilizing effect in naïve cells. In fact, we now show that the knockdown of linear RERE isoforms has no effect on miR-128-3p activity in luciferase assays (**new suppl. Fig. 11A**), strongly arguing against a role for endogenously expressed linear isoforms. Third, expressing a more natural, protein-encoding full-length linear RERE variant is probably also not an option, since the miR-128-3p binding sites in this case are within a coding sequence and could be "shielded" by ribosomes engaged in translation.

Even if our rescue experiments described in Fig. 7 C, D do not provide a conclusive proof for the requirement of RERE circularity, they clearly show that miR-128-3p binding sites present within the major circRERE isoforms are required for rescuing

the synapse phenotype caused by circRERE knockdown. We have adapted our conclusions in the main text accordingly.

MINOR COMMENTS:

1. The word “process(es)-enriched” and not “dendritically-enriched” should be used throughout the results section. “Dendritic compartment” should not be used throughout.

We have now used the word “process-enriched” throughout the manuscript.

2. The authors refer to “three structurally similar isoforms (circRERE1, circRERE2 and circRERE3)”. How do the authors define structurally similar isoform? For instance, one could argue that other isoforms are also structurally similar e.g. 5-9 or 4-10/11/12 etc.

We apologize for the ambiguous wording and have now omitted the term “structurally similar” from the manuscript, given that we do not have any information about the structure of these circRNAs.

3. Fig. 1 - there is virtually no information about the methods used to profile circRNAs and this should be added in the results (sequencing depth and read length) even though the RNAseq is derived from Colameo et al., 2021. How can the authors confidently determine the full length of the circRNAs knowing they do not employ nanopore long-read sequencing? This information should be added in the text of the result section.

As requested by this reviewer, we now provide additional details about sequencing depth and read length in the materials and methods section. More details about methods used to profile circRNA can be found here: Cheng et al., 2016 (PMID: 26556385).

The reviewer is right that the exact sequence of a circRNA cannot be determined without long-read sequencing. However, the presence of different circRERE isoforms could be confidently assessed based on the presence of unique back-splicing junctions, which also matched sequences in public circRNA repositories. In addition, we were able to infer the complete exon inclusions in the main circRERE isoform, circRERE-2, based on the size of DNA fragments obtained by rolling-circle amplification (suppl. Fig. 4B).

4. Fig. 2A-B - PSD95 and Syn1 are difficult to distinguish in the figure – perhaps use different colours? It’s not straightforward to understand how synapse density was quantified and this should be clarified.

We have now attempted to make it easier to distinguish PSD95 (green) and Syn1 (blue) by providing high magnification images of dendritic segments, using arrows to indicate the occurrence of PSD95/Syn1 co-clusters (Fig. 2B, middle panel). This scheme has been consistently used throughout all figures which show synapse co-cluster data. With regard to the quantification, we provide further details on the script used for automated analysis in the methods section. Furthermore, we make the python script, which can be

used as a Fiji plug-in, publically available (<https://github.com/dcolam/Cluster-Analysis-Plugin>).

5.Fig. 3D - the author writes: “Over the course of rat hippocampal neurons differentiation in vitro, both linear and circular RERE isoforms initially decline, but peak at times of synapse formation (15-25 DIV; Fig. 3D).” To be able to make this claim, the authors would need to present replicates and appropriate statistical analysis (or remove this dataset). Here, it seems that only linRERE peaks. What is the circRNA to the linear ratio of circRERE isoforms throughout the course of synaptogenesis here?

As suggested by this reviewer, we have now removed this dataset due to the lack of biological replicates.

6.Fig. 4I - it is unclear why negative values are obtained for GluA1 quantification, since they represent the number of puncta inside a specific area.

We apologize for this mistake and have now fixed this issue in the revised manuscript (new Fig. 4I)

7.Fig5D - what does the scale represent?

We have now provided a more detailed explanation of scales associated with the different curves in the CD plot in the respective figure legend.

8.Fig. 6D - the figure is not representative of the data shown in Fig6E. Pictures illustrating the data should be presented. Furthermore, it seems that neurons are overlapping giving the impression that a signal is present in a cell while it may be derived from a subjacent one (see also comment 6.1 above). How did the authors deal with these case scenarios to ensure that the quantification per cell was accurate?

We have now added more representative pictures in Fig. 6D which better reflect the quantification shown in Fig. 6E. As already pointed out above, our confocal microscopy approach allows us to unambiguously assign puncta to specific cells and compartments. We have shown an example confocal stack for the circRERE-2 smFISH to illustrate this (new suppl. Fig. 5).

9.Fig. 6G (and Fig2B, Fig3F, 4H, 6D, 6G, 7C) - the images illustrating change in dendritic spine densities are not all at the same scale which prevents a visual comparison. For each panel, zoomed pictures should be cohesively shown at the same scale to be able to compare them.

We have made every effort to provide comparable representative pictures in the revised paper. In addition, we want to point out that scale bars have been added to all microscopy pictures, which allows a direct comparison between different panels.

10.Fig. 6H/S3A/B - Statements should be backed up by statistical analysis. For instance related to Fig6H: “Interestingly, miR-128OE alone did not further decrease synapse density, possibly due to its high expression already at basal levels.” is not assessed by statistical analysis comparing adequate groups. Similarly related to suppl. Fig. 3 A/B, the authors write: “ we found that chronic Ptx treatment changed the expression of a distinct subset of circRNAs (circular-to-linear ratio, FDR<0.1) (suppl. Fig. 3 A/B) in a compartment-specific manner, e.g., circHomer1 (You et al., 2015) and several isoforms of the previously characterized circSlc8a1 (Hanan et al., 2020)”. This is not clearly shown by the data (e.g. there is no change in circHomer1 and circSlc8a1 (isoform 3) expression in the dendrites upon Ptx application). This statement should be toned down or backed up by clear statistical analysis.

Since it was also mentioned by reviewer 2 that our PTX dataset didn't lead to clear conclusions, we decided to remove the original Suppl. Fig. 3 entirely. This does not affect any of our claims regarding the function of specific circRNAs in synapse formation and their mechanisms of action.

11.Suppl. Fig1 - the circularity of circRERE should also be validated with RNase R treatment.

We want to politely indicate that the requested experiment had already been included in the original version of the manuscript (suppl. Fig. 1B).

12.Suppl Fig4 - the outcome of the statistical analysis is missing on all panels. p-values should be shown.

We report all the respective p-values in Supplementary Table 3.

13.Suppl Fig5B, vs Suppl Fig6B - is the linear counterpart of circRERE isoforms linRERE or linRERE2? These two supplementary figures do not agree.

We apologize for this mistake and have now used the term “linRERE” for the linear RERE mRNA throughout the paper.

14.Fig. S5D - do circRERE1 and 3 have a similar distribution as circRERE2? Does ISH reveal an enrichment in processes? Similarly in suppl. Fig. 5B, does rolling circle amplification demonstrate the circularity of circRERE1 and 3?

The circularity of circRERE isoforms 1-3 has been confirmed by published nanopore full read sequencing which also provides internal sequence information (Wu et al., NAR 2024; <https://doi.org/10.1093/nar/gkad770>). As discussed above, both circRERE-1 and -3 are also enriched in processes based on our RNA-seq results (Fig. 3B), however, their absolute expression is much lower (in particular circRERE-3) compared to circRERE-2. Given that smFISH for circRERE-2 was already extremely challenging and required extensive optimization, we did not pursue any additional smFISH experiments for the low abundant circRERE-1 and -3.

15. Suppl. Fig. 5E - the scale is not visible and it is not clear which RERE isoforms does the heat-map correspond to.

We have now improved the resolution of the entire figure panel (new Suppl. Fig. 4C) and indicated which human RERE isoforms correspond to rat circRERE 1 and 2 in the respective figure legend.

16. Suppl Fig 6A - the strategy employed to KD circRERE (circRERE1,2,3 isoforms) with shRNA is unclear. They do mention that they “simultaneously [knock] down circRERE isoforms 1-3” but which shRNA pool was selected from the results shown in Suppl Fig. 6A is not described and should be added in the MS. This is particularly relevant because 3 shRNA for circRERE2 is tested in Suppl Fig. 6. A schematic illustrating the shRNA used to KD circRERE 1 and 3 should also be added in Fig S6A.

We have further improved our graphical description of the different shRNAs targeting specific circRERE isoforms in the new suppl. Fig. 6D.

17. The authors should consider using the correct nomenclature for circRERE1, 2, 3, the first time they introduce them in the text of the results (See Chen et al., 2021).

According to this suggestion, we have now indicated the correct nomenclature of the circRERE isoforms on the occasion of their first appearance in the text.

18. Discussion: the authors write: “We note however that absolute quantification of miR-128-3p by qPCR in primary hippocampal neurons led to an estimate of about ~4000 copies of miR-128-3p per neuron (de la Mata et al., 2015), which suggests that miRNA FISH is capturing only a subset of the total population (Fig. 6D-F). Thus, miR-128-3p is presumably present at sufficient quantity in dendrites to control the local expression of synaptic target mRNAs.” I find this statement really far-fetched. I don’t think the authors can strictly conclude about the sensitivity of their assay without experimentally determining copy number. The authors should be really cautious with their statement here and possibly remove it.

We agree with this statement and have toned down our statements in the text accordingly.

TYPOS:

1. Fig. 1F – x-axis is missing in one of the two graphs.

We have added the missing x-axis in this graph.

Reviewer #2 (Remarks to the Author):

The manuscript by Kelly, Schratt and colleagues presents evidence for a functional role of circRNAs produced by back-splicing of the RERE gene transcripts in synaptogenesis. The study is comprehensive and the paper is well written. Below are suggestions for improvement.

Major

1. Page 6. “Together, our bioinformatics results support a role for process-enriched circRNAs in synaptogenesis and synaptopathies, such as Autism Spectrum Disorder”. This is an overstatement, please rephrase. The gene ontology enrichment data shows that genes encoding synaptic proteins are over-represented among those generating process-enriched circRNAs. These data do not demonstrate a role of process-enriched circRNAs in synaptogenesis (the rest of the paper does so for circRERE), and definitely does not provide evidence for their role in ASD.

We apologize for the overstatement and have rephrased this sentence to “Together, our bioinformatics results are consistent with a role for process-enriched circRNAs in synaptogenesis and indicate a potentially exciting link to synaptopathies, such as autism spectrum disorders.”

2. Please add information on the statistical test for each p-value and exact p-value. eg Page 6: “miRNA binding sites are slightly more abundant in dendritically enriched circRNAs compared to the total population ($P < 0.044$)” What test? What is the exact p-value?

We have now added exact p-values and tests to the figure legends of suppl. Fig. 2.

3. The first section of the paper is quite long, with some of the results rather vague or weak. I'd suggest removing some of the data that does not lead to clear conclusions such as: “We found that chronic Ptx treatment changed the expression of a distinct subset of circRNAs (circular-to-linear ratio, $FDR < 0.1$) (suppl. Fig. 3 A/B) in a compartment-specific manner, e.g., circHomer1 (You et al., 2015) and several isoforms of the previously characterized circSlc8a1 (Hanan et al., 2020). However we observe distinctive expression of circRNA BSJs (suppl. Fig. 3C) and the total exons of the circRNA tied to linear+circular counts, requiring further investigation to characterise their dynamic expression profiles (suppl. Fig. 3D).”

Please see our comment to reviewer 1, Minor comment 10

4. SFig3 lacks pvalues

Suppl. Fig. 3 has now been removed (see our comment above).

5. Why does the siControlPool show significant downregulation in synapse density vs siControl1 in Fig2C? I would expect it to show no significant difference if there were no off-target effects.

Please see our comment to reviewer 1, Major comment 7.4.

6. Please add quantification to Fig 3E as the image on its own is not clear enough.

We have now performed more specific circRNA FISH using a BSJ-targeting probe and have replaced the image accordingly (**new Fig. 3D, E**). Please see also our comment to reviewer 1, Major comment 5.

7. Page 10. "However, the knockdown of all 3 circRERE isoforms resulted in a more robust increase in dendritic synapse density compared to the circRERE2 shRNA (Figure 3G)" - there is no statistical test supporting this statement, so it's not clear if the difference is significant. Same comment for Fig4I and the corresponding text.

We agree with the comment of this reviewer and have now carefully re-phrased our statements regarding the function of different circRERE isoforms. In essence, all our results are consistent with a predominant role of circRERE-2, and some contribution of circRERE-1, but not -3, to the observed phenotypes. Please see also our comments to reviewer 1, Major comment 3.1.

Minor

1. The labelling of Fig2C y-axis is confusing. Perhaps explain in the legend that it is an inverse log-scaled axis. Or else, just use $-\log_{10}(\text{p-value})$ which is common for Volcano plots.

We have now changed the y-axis scale to $-\log_{10}(\text{p-value})$ as requested.

2. Fig 5A please add the pvalue and FC thresholds to the legend.

We have added the respective cutoffs to the figure legend of Fig. 5A

Reviewer #3 (Remarks to the Author):

Dear Editor and Authors,

I appreciate the opportunity to review the manuscript titled "A functional screen uncovers circular RNAs regulating excitatory synaptogenesis in hippocampal neurons" by Darren Kelly et al. This paper presents a systematic investigation into the role of circular RNAs (circRNAs) in synaptogenesis. Circular RNAs have emerged as crucial regulators of gene expression, and their roles in neural development and function are becoming increasingly recognized as highly relevant in the field of neurobiology. Utilizing RNA interference, the authors identified several circRNAs that act as negative regulators of synapse formation and focus on circRERE, thus contributing to our understanding of circRNA-dependent microRNA regulation in synapse development and function. The study is well-executed, with robust experimental design and insightful discussion, and it gives a clear model for the proposed mechanism, making a significant contribution.

While I am very positive about the manuscript and its contribution to the field, I request some

clarifications and additional details to enhance the clarity and comprehensiveness of the study.

Clarification on the Use of Different Neuron Types: On page 10, the manuscript mentions using rat cortical neurons for some experiments despite primarily focusing on hippocampal neurons. No further explanation is provided for this switch. Could the authors clarify the rationale behind using cortical neurons in these experiments?

We had to use rat primary cortical neurons for a small number of biochemical assays (e.g., circRERE knockdown validation by qPCR after nucleofection; suppl. Fig. 6A-F), which require a high number of cells which is not easily achieved with primary hippocampal neurons. However, these experiments served the only purpose to validate the efficiency and specificity of our knockdown constructs, and therefore did not affect any of our conclusions regarding circRERE function in synaptogenesis.

Synapse and Dendritic Spine Analysis: I appreciate that the authors have reported synapse formation using both Syn/PSD95 co-labeling and GFP-labeled dendritic spine analysis. However, the exact numbers should be reported individually for PSD alone, Syn alone, and double-label synapses. This level of detail is essential for a clear understanding of the synapse distribution. Additionally, please clarify the criteria for selecting the counted segments, specifically regarding their distance from the soma, as spine density and GFP area depend highly on this parameter.

We have now added the individual analysis of Synapsin-1 and PSD-95 puncta, as well as dendritic spine density data, to the respective figures in the supplement (**new suppl. Fig. 7 A-C; new suppl. Fig. 10 C-E; new suppl. Fig. 12 A-C**).

With respect to area selection for counting, we want to stress that we didn't pick specific dendritic segments, but rather the entire cell (based on the GFP mask) excluding the nucleus. This in our view is the most unbiased approach, as the reviewer correctly pointed out that synapse co-cluster and dendritic spine parameters are heavily dependent on their location within the neurons.

Number of Dendritic Spines in GluA1 Puncta Analysis: In the figure reporting the number of GluA1 puncta, the neurons are also labeled with GFP. I suggest including the number of dendritic spines in this same set of data.

We have now performed the analysis of dendritic spine density on two GluA1 datasets, one originating from the previous live immunostaining (GluA1 surface expression; new Fig. 4H, I) and new data derived from GluA1 immunostaining on fixed cells (total GluA1 expression; new Fig. 4J, K). Consistent with our previous results, we observe a significant increase in dendritic spine density upon circRERE sh expression in fixed cells subjected to GluA1 immunostaining (**new suppl. Fig. 8F**). In contrast, we did not observe significant changes in spine density in samples which underwent live staining (not shown). This can likely be explained by the fact that dendritic spine morphology in general is not well preserved in unfixed samples. The result from this analysis might therefore be misleading, and we decided not to include them in the revised manuscript.

Synapse Number in Figure 6: Please include the synapse number based on GFP-labeled dendritic spine analysis to provide a consistent metric for comparing different figures.

We have now added the dendritic spine density analysis for our miR-128 overexpression experiments (**new suppl. Fig. 10E**).

Figure Legends: The figure legends currently need more detail. Please include more information about the experimental design to help readers understand the data and its conclusions.

We carefully checked all figure legends and added information which is necessary to allow the reader to evaluate our results (e.g, detailed statistics). However, we were unable to add very detailed explanations of the experimental design in each of the figure legends due to space constrains (max. 350 words per legend). We have however included those in the materials and methods sections.

Typo in Figure 7: The term "GLLM" appears to be a typo. If I am correct, it should likely be "GLMM" (Generalized Linear Mixed Models).

Thanks a lot for pointing this out, we have corrected the typo accordingly.

Overall, I am very enthusiastic about this manuscript. It addresses a highly relevant topic and is of excellent quality. With these minor revisions, I strongly recommend its publication.

Thank you for considering these suggestions. Addressing these points will further strengthen the manuscript and provide additional clarity for readers.

Q1: “We observed that ADAM10 inhibition led to a reduction in synapse co-clusters under control conditions and normalized synapse co-cluster density under circRERE KD conditions.” Here the data is not included in the reply to reviewers.

R1: we apologize for this mistake and have now added the requested data below. It shows that in circRERE KD neurons, pharmacological ADAM10 inhibition by GI254023X significantly reduces synapse co-cluster density, bringing it back to baseline levels.

contrast	Transfection	estimate	SE	df	t.ratio	p.value
GI 10uM - DMSO	ScramSh	-0.065824888	0.12181583	230.006764	-0.540363983	0.589466
GI 10uM - DMSO	circREREKd	-0.414888571	0.12027223	230.000007	-3.449579085	0.000667

Q2.1: “Using a miR-128-3p perfect binding site luciferase reporter assay (Fig. 6C), we found that knockdown of both circRERE-1 and 2, but not-3 led to a significant decrease in miR-128-3p activity.” Such data is now shown in Fig. 6C.

R2.1: We apologize for the mis-labeling of figures. As pointed out by the reviewer, this data can actually be found in **suppl. Fig. 10A**. We have changed the references in the manuscript accordingly.

Q3.1. “we consistently observe a trend towards bigger effect sizes with the circRERE shRNA pool targeting all 3 major circRERE isoforms compared to circRERE-2 alone (e.g., synapse density (Fig. 3G), GluA surface expression (Fig. 4H),

miR-128-3p activity (Fig. 6C)),” Fig. 4H and 6C does not compare circRERE shRNA with circRERE-2 shRNA alone.

R3.1: We apologize for omitting the circRERE-2 shRNA data in the revised version and have now added it to Fig. 4I and 4K. With regard to Fig. 6C, please see our response R2.1 above.